# Synthesis and preliminary PET imaging of [11]C and [18]F isotopologues of the ROS1/ALK inhibitor lorlatinib

Thomas Lee Collier[1,2], Marc D. Normandin[1], Nickeisha A. Stephenson[1,†], Eli Livni[1], Steven H. Liang[1], Dustin W. Wooten[1], Shadi A. Esfahani[1], Michael G. Stabin[3], Umar Mahmood[1], Jianqing Chen[4], Wei Wang[5], Kevin Maresca[4], Rikki N. Waterhouse[4,†], Georges El Fakhri[1], Paul Richardson[5] & Neil Vasdev[1]

Lorlatinib (PF-06463922) is a next-generation small-molecule inhibitor of the orphan receptor tyrosine kinase *c-ros oncogene 1* (ROS1), which has a kinase domain that is physiologically related to anaplastic lymphoma kinase (ALK), and is undergoing Phase I/II clinical trial investigations for non-small cell lung cancers. An early goal is to measure the concentrations of this drug in brain tumour lesions of lung cancer patients, as penetration of the blood–brain barrier is important for optimal therapeutic outcomes. Here we prepare both [11]C- and [18]F-isotopologues of lorlatinib to determine the biodistribution and whole-body dosimetry assessments by positron emission tomography (PET). Non-traditional radiolabelling strategies are employed to enable an automated multistep [11]C-labelling process and an iodonium ylide-based radiofluorination. Carbon-11-labelled lorlatinib is routinely prepared with good radiochemical yields and shows reasonable tumour uptake in rodents. PET imaging in non-human primates confirms that this radiotracer has high brain permeability.

[1] Massachusetts General Hospital and Harvard Medical School, 55 Fruit Street, Boston, Massachusetts 02114, USA. [2] Advion Inc., 30 Brown Road, Ithaca, New York 14850, USA. [3] Vanderbilt University, 1161 21st Avenue South, Nashville, Tennessee 37232, USA. [4] Pfizer Inc., Quantitative Medicine, Worldwide Research and Development, 1 Portland Street, Cambridge, Massachusetts 02139, USA. [5] Pfizer Worldwide Research and Development, 10770 Science Center Drive, San Diego, California 92121, USA. † Present addresses: Department of Chemistry, University of the West Indies, Mona, Kingston 7, Jamaica (N.A.S.); Waterhouse Imaging and Biomarker Consultants, 208 Candia Road, Chester, New Hampshire 03036, USA (R.N.W.). Correspondence and requests for materials should be addressed to N.V. (email: vasdev.neil@mgh.harvard.edu).

Anaplastic lymphoma kinase (ALK) is a member of the insulin receptor family of receptor tyrosine kinases, and is primarily expressed in the central nervous system[1,2]. Over a dozen ALK fusion partners have been identified in several cancer types[1–4], and this has ultimately led to clinical translation and United States Food and Drug Administration approval of ALK inhibitors[5] as a first-line treatment for ALK-positive lung cancer patients. Lorlatinib (PF-06463922) is a next-generation, small-molecule inhibitor of the orphan receptor tyrosine kinase *c-ros oncogene 1* (ROS1), which has a kinase domain that is physiologically related to and also inhibits ALK[6]. This orally available, ATP-competitive inhibitor has shown excellent therapeutic potential against ROS1-driven fusion cancers, and significantly improved inhibitory activity compared with the first-generation-approved tyrosine kinase inhibitors (TKIs) including the ALK/mesenchymal–epithelial transition factor/ROS inhibitor, crizotinib (Xalkori)[7], as well as next-generation ALK and ALK/ROS1 inhibitors, ceritinib and alectinib[8,9]. Most ALK inhibitors (and all known TKIs) are limited by acquired resistance to therapy, generally driven by mutations that alter the kinase domain of ALK, activation of other oncogenic signals or pharmacokinetic (PK) issues with the drug and are generally not optimized for brain penetration[10]. A common site of metastases in non-small cell lung cancer (NSCLC) patients is in the brain where previous generation ALK inhibitors have limited effectiveness, and may be attributed to poor blood–brain barrier (BBB) penetration or active transport out of the brain by efflux pumps[11–13]. Lorlatinib was specifically designed to address this unmet clinical need for robust brain penetration and activity against TKI-resistant ALK mutants, including the crizotinib-, alectinib- and ceritinib-resistant fusion of echinoderm microtubule-associated protein-like 4 (*EML-4*)-*ALK* Gly1202Arg mutant[14]. This drug is presently undergoing Phase I/II clinical trial investigations (http://clinicaltrials.gov/ct2/show/NCT01970865) in ROS1 and ALK fusion-positive NSCLC patients[13,15–21].

Unfortunately, imaging receptor tyrosine kinases and their associated signal transduction pathways with isotopologues of potent and selective drugs are rarely explored for oncology and neuroimaging[22–27] with positron emission tomography (PET). This lack of effort is partially attributed to the additional challenges of imaging intracellular targets, compared to high-density receptor and enzyme targets at the extracellular domain, and competition at binding sites with high intracellular levels of ATP. These difficulties are further exacerbated by the challenging radiochemistry required to prepare isotopologues of the structurally complex potent and selective TKIs. A PET radiotracer for ALK could aid many ongoing clinical trials with ALK-targeted therapeutics by indicating the success and extent of engagement by ALK in the periphery and central nervous system, as well as for occupancy studies to assess dose–response and to understand the PK properties of a labelled drug. Although this effort has been hindered by the low BBB permeability of crizotinib[28] and related compounds, there is a critical need to develop a PET neuroimaging agent for ALK; hence, our goal is to evaluate the PK of lorlatinib by PET for clinical oncology. Such a radiotracer would enable us to further our understanding of ALK–drug concentrations in normal brain and in brain metastases. In the present work, we synthesize carbon-11 ($^{11}$C; $\beta^+$, $t_{1/2} = 20.4$ min) and fluorine-18 ($^{18}$F; $\beta^+$, $t_{1/2} = 109.7$ min)-labelled isotopologues of lorlatinib and carry out preliminary biodistribution and PET imaging using [$^{11}$C]lorlatinib in tumour-bearing rodents and non-human primates (NHPs).

## Results

**Precursor synthesis.** With the goal of exploring both $^{11}$C and $^{18}$F radiolabelling of lorlatinib, ((10*R*) − 7-amino-12-fluoro-2,10,

16-trimethyl-15-oxo-10,15,16,17-tetrahydro-2*H* − 8,4-(metheno) pyrazolo[4,3 − *h*][2,5,11]-benzoxadiazacyclotetradecine-3-carbonitrile (PF-06463922; (*R*)-1), five compounds ((*R*)-2 − (*R*)-6; Fig. 1) were synthesized to serve as potential precursors. Compounds (*R*)-2 and (*R*)-3 were intended to be evaluated for $^{11}$C labelling through methylation of the amide nitrogen, whereas compound (*R*)-6 was a desirable intermediate to assess an iodonium ylide labelling strategy[29–31]. Compounds (*R*)-4 and (*R*)-5 were attractive from the perspective of utilizing nucleophilic aromatic substitution approach to access the $^{18}$F-labelled material.

**Synthesis of labelling precursors for lorlatinib ((*R*)-1).** Compound (*R*)-2 was originally synthesized as a standard for metabolite studies and also serves as a precursor for $^{11}$C labelling with [$^{11}$C]CH$_3$I (Fig. 2; Supplementary Fig. 2). With compound (*R*)-2 in hand, it may be possible to selectively methylate the amide nitrogen with [$^{11}$C]CH$_3$I; however, a more likely labelling strategy would involve protection of the aminopyridine motif, and carry out a two-step alkylation/deprotection sequence. Attempting to simply selectively *bis*-Boc-protect the aminopyridine compound (*R*)-2, led under standard conditions, to mainly the *tri*-Boc-protected compound, and all attempts to selectively remove the Boc group from the amide nitrogen failed. With these results, and a limited amount of materials in hand, an alternative approach to (*R*)-3 was devised. As we have previously reported[6], a direct arylation can be utilized to form the macrocyclic ring, although the success of this approach is to some degree enabled by protection of the amino pyridine[6]. This is reinforced by a poor yield (<5%) being obtained when direct cyclization was attempted to form compound (*R*)-2. However, this approach is successful for (*R*)-3 (Fig. 3; Supplementary Fig. 3).

**Synthesis of [$^{11}$C]Lorlatinib ([$^{11}$C](*R*)-1).** Owing to the short half-life of carbon-11, our initial attempts to synthesize [$^{11}$C](*R*)-1 employed a one-step direct methylation of unprotected precursor (*R*)-2 with [$^{11}$C]CH$_3$I using the 'Loop Method'[32] (Fig. 2). In this established method, the entire reaction is carried out on an high-performance liquid chromatography (HPLC) loop prior to purification, and no losses to transfer of reagents occur. The 'Loop Method' is commonly used instead of vial-based $^{11}$C-methylation reactions, because of its simplicity (no heating or cooling required) as well as ease of automation, including adaptation to the commercial radiosynthesis modules. Labelling of precursor (*R*)-2 was initiated by flowing [$^{11}$C]CH$_3$I through an HPLC loop coated with a thin film of (*R*)-2 (1 mg) in *N*,*N*-dimethylformamide (DMF) with 0.7 equivalents of tetrabutylammonium hydroxide (TBAOH, 1 M in methanol) at room temperature. After 5 min, the contents of the HPLC loop were eluted with 10% water in acetonitrile to give 25% uncorrected radiochemical yield (RCY) of a labelled compound. Analysis of the crude reaction mixture by HPLC indicated that the major radiolabelled product had a similar retention time as the desired product, but was not easily separated from the precursor (*R*)-2, using semipreparative HPLC. It cannot be ruled out that the major radiolabelled product in this reaction was not the desired product, [$^{11}$C](*R*)-1, but was the methyl pyridinium salt [$^{11}$C](*R*)-7. Methylation of 2-aminopyridines occurs favourably at the pyridine nitrogen because of resonance electron-releasing properties of the amino group, which leads to enhanced nucleophilicity of the pyridine[33]. In light of the challenge to isolate [$^{11}$C](*R*)-1 from the reaction mixture with confidence and suitable radiochemical purity for clinical translation, we abandoned the 1-step strategy and focused on a more robust 2-step $^{11}$C-labelling process that employs a diBoc-protected amine for routine radiotracer production.

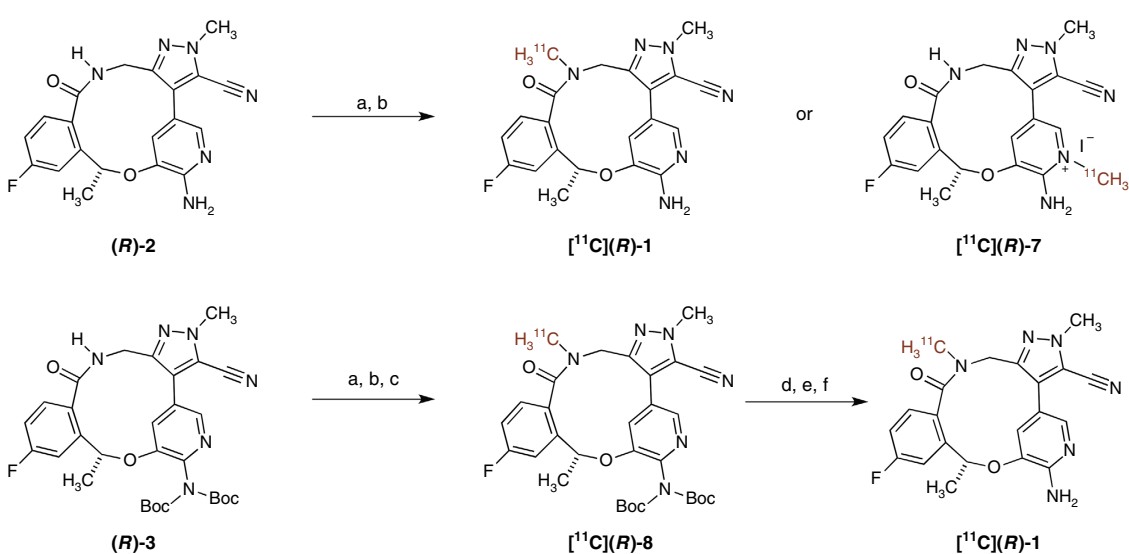

Lorlatinib (PF-06463922; **(R)-1)**)

**(R)-2**          **(R)-3**

¹¹C Precursors

**(R)-4**          **(R)-5**          **(R)-6**

¹⁸F Precursors

**Figure 1 | Structures of lorlatinib and labelling precursors.** The target compound lorlatinib is shown at the top. Sites of derivatization are shown in blue for reactions with [¹¹C]CH₃I (middle) or green for [¹⁸F]fluoride (bottom). The iodonium ylide precursor was prepared from **(R)-6**.

**(R)-2** —a, b→ [¹¹C](**R**)-1   or   [¹¹C](**R**)-7

**(R)-3** —a, b, c→ [¹¹C](**R**)-8 —d, e, f→ [¹¹C](**R**)-1

**Figure 2 | Development of [¹¹C]lorlatinib.** The one-step reaction (top) presented challenging separation of the precursor **(R)-2** from [¹¹C]lorlatinib and confirmation of identity, which is likely attributed to **[¹¹C](R)-7**. A two-step methodology (bottom) allowed for facile removal of the precursor and simplified the automation for routine production. Reagents and conditions: (a) 1 mg of precursor (**(R)-2** or **(R)-3**), 0.7 eq TBAOH (1 M in methanol), in 80 µl anhydrous DMF; (b) [¹¹C]CH₃I, via 'Loop Method'; (c) HPLC purification (see ESI); (d) 0.7 ml of 6 M HCl, 80 °C, 5 min; (e) 4.5 ml of 1 N NaOH, 6 ml of 3 M NaOAc, 6 ml H₂O; (f) SPE reformulation.

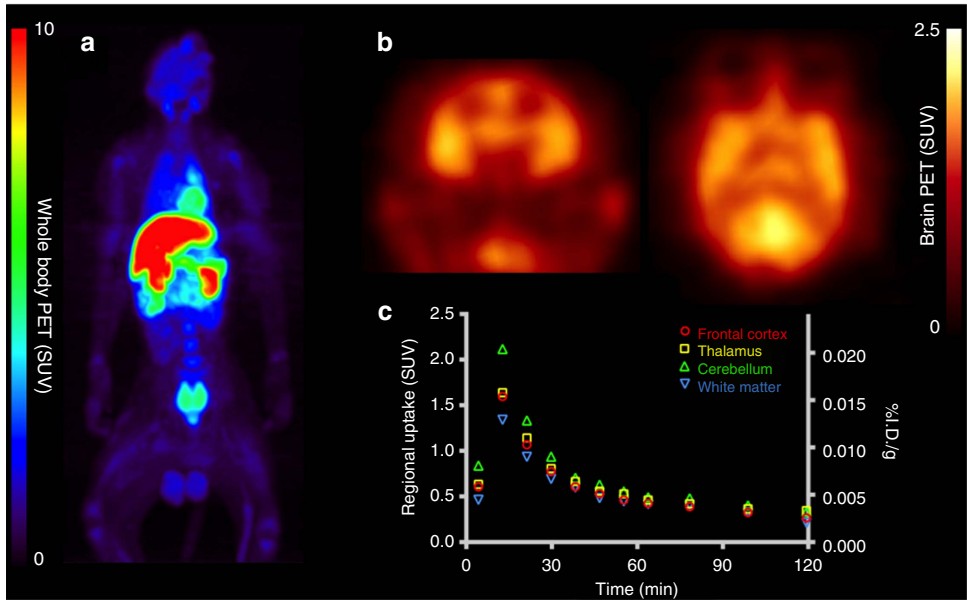

**Figure 3 | Non-human primate PET imaging of [$^{11}$C]lorlatinib.** Representative dynamic multibed PET imaging data with [$^{11}$C]**(R)-1** in rhesus macaque. (**a**) Maximum intensity projection PET images from the first two whole-body passes corresponding to data acquired over the initial 17 min after radiotracer administration. (**b**) Axial and coronal slices of PET images at peak-measured brain levels corresponding to data acquired at ~10 min post injection. (**c**) Regional time activity curves exhibit rapid uptake, followed by fast washout of the radiotracer.

In order to attenuate the reactivity of the pyridine and to favour selective labelling of the amide functionality, a doubly Boc-protected molecule, precursor **(R)-3**, was used in a two-step labelling employing [$^{11}$C]CH$_3$I via the aforementioned 'Loop Method', followed by purification and hydrolysis on a modified commercial radiofluorination unit (GE TracerLab FX$_{FN}$; Supplementary Fig. 13). By using the two-step methylation procedure (Fig. 2), the labelled intermediate [$^{11}$C]**(R)-8** could be easily separated from the precursor **(R)-3**, by semipreparative HPLC, and could then be rapidly deprotected and purified (Supplementary Fig. 14). A slow stream of [$^{11}$C]CH$_3$I in helium was pushed into an HPLC loop containing **(R)-3,** TBAOH and DMF over 4 min. The mixture was allowed to react in the loop, and after 5 min the solution was injected directly into the HPLC system for semipreparative purification. The large difference in the chromatographic properties of protected labelled product [$^{11}$C]**(R)-8** allowed for the complete separation of **(R)-8** from the starting precursor **(R)-3**. The desired radiolabelled product, [$^{11}$C]**(R)-8**, was collected by HPLC separation and the Boc groups were deprotected with HCl (6 N) in ethanol at 80 °C for 5 min. The resulting solution was cooled, and then neutralized with NaOH (1 N) and buffered with sodium acetate (3 M). The final product was purified by solid phase extraction (SPE) and reformulated in 10% ethanol in saline to provide [$^{11}$C]lorlatinib ([$^{11}$C]**(R)-1**), in 3% uncorrected RCY (17% decay-corrected) at end of synthesis (50 min) and a high specific activity of 3 Ci µmol$^{-1}$, with >95% radiochemical purity.

This work demonstrates the proof of concept labelling of a two-step $^{11}$C reaction that can be obtained on a commercial radiofluorination unit with a rapid and simple reversible modification that does not alter the standard plumbing of the GE TracerLab FX$_{FN}$. We anticipate that this methodology may find applications in other $^{11}$C-labelling sequences involving two-step reactions[34,35].

***In vivo*** **PET imaging of [$^{11}$C](R)-1.** PET imaging studies were initially performed with [$^{11}$C]**(R)-1** in NHPs. In each of two rhesus macaques, high initial uptake was seen in the brain after intravenous injection of [$^{11}$C]**(R)-1**, and is consistent with the previously reported brain penetration properties of the non-radioactive molecule[6]. Whole-body dynamic PET imaging revealed radioactivity primarily in the liver and kidneys (Fig. 3a), the latter exhibiting signal primarily at early time points that progressively shifted to the urinary bladder (Supplementary Figs 22 and 23). The peak-measured brain concentrations were locally high with the cerebellum exceeding a standardized uptake value (SUV) of 2 at ~10 min post injection (Fig. 3b); the maximum measured per cent injected dose in the brain was 1.4% at ~10 min post injection. SUV time activity curves exhibit rapid uptake, followed by fast washout of the radiotracer in normal NHPs (Fig. 3c). Regional uptake exhibits modest heterogeneity but is generally concordant with expected ALK distribution, with highest [$^{11}$C]lorlatinib concentrations in the cerebellum, frontal cortex and thalamus; intermediate levels in other cortical grey matter; and lowest values in the white matter[36].

It is noteworthy that a preliminary assessment of tumour uptake of this radiotracer was also carried out in mice bearing subcutaneous human H3122 (EML4-ALK-positive) NSCLC xenografts and evaluated by PET-CT imaging in conjunction with blocking studies (Supplementary Figs 24 and 25). Briefly, our initial results show that the tumour uptake reached its plateau in ~30–60 min after injection of [$^{11}$C]**(R)-1** (with 2.2–2.37% ID/g) and co-injection with **(R)-1** resulted in a significant decrease in the tumour uptake (<0.4% ID/g) during the entire imaging course of 90 min.

The dosimetry of the $^{11}$C-tracer was determined in NHPs after the injection of [$^{11}$C]**(R)-1**. Doses from any carbon-11 compound are generally low, because of the short physical half-life[37]. Radioactivity levels were measured in various organs of the animals at 13 time points at up to ~120 min post injection. With these data, the liver dose appears to be 0.021 mSv MBq$^{-1}$, and the effective dose appears to be 0.0035 mSv MBq$^{-1}$ (Supplementary Table 1). The dosimetry estimates from NHPs are consistent with those of other $^{11}$C-labelled compounds. Clinical translation of this radiotracer is warranted and regulatory

submissions should be facilitated, given that further toxicity studies are not required.

**Synthesis of labelling precursors for [18F]lorlatinib.** The longer half-life of fluorine-18 offers many advantages for radiosynthesis and imaging protocols compared to carbon-11. Specifically, the ability to conduct several scans from a single production of an [18F]-labelled radiotracer, facilitation of widespread use and multicentre trials, as well as longer scanning protocols can be achieved. Therefore, [18F]-labelling is a preferred option, particularly since a direct isotopologue of lorlatinib could be prepared. For the corresponding nitro-aromatic precursors needed for [18F]-labelling (**4** and **5**; Supplementary Fig. 7), both the standard amidation and the direct arylation approaches were also investigated to access this novel core. Rather than start with enantiopure starting material, we pursued a less labour-intensive late-stage chiral separation, and then correlated the product to the desired enantiomer.

Our current synthetic methodology was useful in securing enough substrate for initial labelling investigations; however, in order to provide a sustainable supply of key intermediates, a higher-yielding methodology was needed. Synthesis of cores such as (**R**)-**4**, (**R**)-**5** and potentially (**R**)-**6** became key to the success of the project. A logistical challenge was to determine at what point to carry out the chiral separation. Having correlated the stereochemistry of the chiral alcohol (**S**)-**19** to the final macrocycles, we decided that it would be more facile in terms of scale to carry out the separation on either **4** or **5** with the final decision being based on which displayed better solubility and more efficient chiral separation. In addition, investigating a direct arylation route to these compounds could potentially lead to an improvement in yield, avoidance of issues with nitrile hydrolysis as well as a shortening in the synthetic route (Fig. 4).

Using the syntheses outlined in Fig. 4, multigram quantities of racemic **5** could be obtained, which could then be separated by chiral supercritical fluid chromatography (SFC). Correlation of the materials obtained with the compound obtained from (**S**)-**19** enabled the isolation of (**R**)-**5** on a gram scale.

The harsh conditions required for fluorodenitration of both (**R**)-**4** and (**R**)-**5** lead to significant decomposition of the macrocycle, in particular with respect to the sensitivity of the

nitrile moiety (*vide infra*). As such, alternative approaches to [18F]-labelling were also investigated. We recently reported on a spirocyclic hypervalent iodine(III)-mediated radiofluorination strategy, based on iodonium ylides to afford [18F]-aryl fluorides in high RCYs[29]. The technique involves stable, easily purified precursors and is readily implemented with standard workup procedures. The conceptual advantages of excellent regioselectivity and viability of incorporation of [18F] into a wide array of non-activated (hetero) arenes make this methodology suitable for use on non-activated aromatic rings. With this methodology in mind, access to (**R**)-**6** became attractive, and we were confident that this compound could be accessed from the enantiopure (**R**)-**5**. As shown in Fig. 5, Boc-protection again provides (**R**)-**4**, which could be reduced to (**R**)-**12** using sodium dithionite. Diazotization/iodination was then utilized to install the requisite iodide to form (**R**)-**6**.

The corresponding ylide (**R**)-**13** was prepared following the general literature procedure[29] (Supplementary Fig. 10); in short, the aryl iodide (**R**)-**6** was reacted with Selectfluor and trimethylsilyl acetate to form the intermediate iododiacetoxy compound. A basic solution of 6,10-dioxaspiro[4.5]decane-7, 9-dione was added to the iododiacetoxy intermediate and the reaction allowed to proceed until full conversion of the iododiacetoxy starting materials. After purification, the iodonium ylide (**R**)-**13** was isolated in 36% yield.

**Radiosynthesis of [18F]lorlatinib.** Attempted manual fluorodenitration with [18F]Et4NF ([18F]TEAF) was performed using the nitro precursors (3–4 mg) in dimethylsulphoxide with heating ranging from 160 to 215 °C for 15 min; however, [18F](**R**)-**1** could not be identified in the reaction mixture. The radiosynthesis was then attempted using the commerical microfluidic system[38]; however, complete HPLC separation of the nitro precursor from the final product proved to be problematic (Supplementary Figs 16 and 17). Given that precursor (**R**)-**4** or (**R**)-**5** possesses an electron-rich aromatic ring and are only mildly activated towards nucleophilic aromatic substitution reactions due to the presence of the *para*-amide, it represents a challenging substrate for labelling with fluorine-18 using conventional labelling methods. It is therefore not surprising that fluorodenitration was not fruitful in our hands by manual or microfluidic approaches. Fortunately,

**Figure 4 | Direct arylation approach to (R)-5 and (R)-6.** Reagents and conditions: (a) 1.5 eq pyrazole **10**, 2 eq T3P (50% in EtOAc), 4 eq TEA, 2-MeTHF, room temperature (rt), 5 h, 79%; (b) 8 mol% CataCXium A, 8 mol% Pd(OAc)₂, 3 eq KOAc, 1 eq H₂O, *t*-AmOH, reflux, 16 h; (c) 4 M HCl/EtOAc, EtOAc, rt, 56 h, 71% over two steps, chiral purification by SFC; (d) 10 eq (Boc)₂O, 0.5 eq DMAP, CH₂Cl₂, 25 °C, 16 h, 76%; (e) 8 eq Na₂S₂O₄, 8 eq NaHCO₃, H₂O/THF (1:1), 20 °C, 16 h, 58%; (f) 2 eq isoamyl nitrite, 2 eq CH₂I₂, CH₃CN, 80 °C, 4 h, 21%.

**Figure 5 | Synthesis of [¹⁸F](R)-1 via an iodonium ylide precursor.** Reagents and conditions: (a) 2 mg iodonium ylide **(R)-13** in 400 µl of DMF (anhydrous), azeotropically dried [¹⁸F]Et₄NF, 80 °C, 10 min.; (b) HPLC purification followed by C18 SPE isolation; (c) 4 M HCl, 90 °C, 10 min; (d) neutralization to pH 5 using NaOH and NaOAc, reformulation on HLB SPE.

our iodonium ylide-based radiochemistry[29–31] proved to be the most effective method for preparation of the ¹⁸F isotopologue of **(R)-1** (Fig. 5), resulting in a 14% uncorrected RCY, relative to starting [¹⁸F]fluoride (compared to the 1% RCY by microfluidic fluorodenitration[39]), and in a radiochemical purity of >97%. We anticipate that this methodology will be applicable for ¹⁸F-labelling of other mildly activated aromatic clinical candidates[40]. This development of [¹⁸F]**(R)-1** is also worthy of further investigations *in vivo*.

## Discussion

Clinically useful amounts of the ¹¹C- and ¹⁸F-isotopologues of the ROS1/ALK inhibitor lorlatinib were synthesized, enabling the possibility for *in vivo* quantification of ALK–drug concentrations and brain metastases. To accommodate the non-traditional radiolabelling strategies that were required, five precursor molecules were synthesized via multistep syntheses and strategically placed chiral separations. Carbon-11- and fluorine-18-labelled lorlatinib were prepared in good RCYs and purities via a unique and fully automated 2-step ¹¹C-labelling strategy, as well as our iodonium ylide-based radiofluorination methodology. The initial PET imaging study was designed to confirm that [¹¹C]lorlatinib readily crosses the BBB. Our future work with [¹¹C]lorlatinib includes further PET imaging in rodent tumour models, normal NHPs, with concurrent automation and preclinical translation of [¹⁸F]lorlatinib.

## Methods

**General chemistry experimental procedures.** Starting materials and other reagents were purchased from commercial suppliers and were used without further purification unless otherwise stated. Lorlatinib, also known as PF-06463922, is now commercially available from Sigma-Aldrich. All reactions were performed under a positive pressure of nitrogen, argon or with a drying tube, at ambient temperature (unless otherwise stated), in anhydrous solvents, unless otherwise indicated. Analytical thin-layer chromatography (TLC) was performed on glass-backed Silica Gel 60_F 254 plates (Analtech (0.25 mm) and eluted with the appropriate solvent ratios (v/v). Flash column chromatography was performed using a Biotage Isolera One system and preloaded Biotage columns. Silica gel for flash chromatography was high-purity grade 40–63 µm pore size and was purchased from Sigma-Aldrich. Reactions were monitored by HPLC or TLC and terminated as judged by the consumption of starting material. The TLC plates were visualized by ultraviolet, phosphomolybdic acid stain or iodine stain. Microwave-assisted reactions were carried out in a Biotage Initiator. All test compounds showed >95% purity as determined by combustion analysis or by HPLC. HPLC conditions were as follows: X-Bridge C18 column at 80 °C, 4.6 mm × 150 mm, 5 µm, 5–95% MeOH/H₂O buffered with 0.2% formic acid/0.4% ammonium formate, 3 min run, flow rate 1.2 ml min⁻¹, ultraviolet detection (λ = 254, 224 nm).

**Spectroscopy.** Multi-NMR spectra were recorded on either a Bruker 300 MHz, a Bruker 400 MHz, or a Varian Unity Inova 500 MHz spectrometer, and resonances are given in parts per million (p.p.m.) relative residual solvent. ¹H NMR spectra were obtained as DMSO-*d₆* or CDCl₃ solutions as indicated (reported in p.p.m.), using chloroform as the reference standard (7.25 p.p.m.) or DMSO-*d₆* (2.50 p.p.m.), unless otherwise stated. Peak multiplicities are reported using the following abbreviations: s = singlet, d = doublet, t = triplet, m = multiplet, br = broadened, dd = doublet of doublets, dt = doublet of triplets. *J* coupling constants, when given, are reported in hertz (Hz). HRMS spectra were recorded on an Agilent 6,220 ESI TOF mass spectrometer using flow injection analysis. Mass

spectrum data were also obtained by liquid chromatography mass spectrometry (LCMS) on an Agilent instrument using atmospheric pressure chemical ionization (APCI) or electrospray ionization (ESI). Selected infrared spectra were recorded from neat compounds on a Bruker ALPHA FT-IR.

**Synthesis of labelling precursors for [¹¹C]lorlatinib.** Compound **(R)-2** was originally synthesized as a standard for metabolite studies and also serves as a precursor for ¹¹C-labelling with [¹¹C]CH₃I (Supplementary Fig. 1). Initial one-pot Suzuki coupling between **(R)-15** and the pyrazole **16**, using B₂pin₂ and CataCXium A as a ligand followed by N-Boc deprotection then ester hydrolysis led to **(R)-19**, the precursor for the macrolactamization. Ring-closure is known to be the challenging step in the formation of macrocycles, and often strategies such as high dilution and slow addition techniques are utilized to address this. The conformational preferences of the precursor also have a strong influence on the success of the ring formation[41], and herein a modest yield was obtained after EDCI-mediated ring formation (it is interesting to contrast this with the yields obtained in the case of the clinical candidate **(R)-2**, which differs by an additional methyl group on the amide nitrogen, which leads to a 10–20% increase in yield for the cyclization step probably due to the greater nucleophilicity of the secondary amine). Other amide-bond-forming reagents were not tested in this step, and the presence of *cis/trans* diastereomers (1/3) around the amide bond can clearly be observed in the NMR spectroscopic and LCMS analysis of **(R)-2**.

**Methyl 2-[(1R)-1-{[2-amino-5-(3-{[bis(tert-butoxycarbonyl)amino]methyl}-5-cyano-1-methyl-1H-pyrazol-4-yl)pyridin-3-yl]oxy}ethyl]-4-fluorobenzoate (R)-17.** To a solution of methyl 2-{(1R)-1-[(2-amino-5-bromopyridin-3-yl)ox-y]ethyl}-4-fluorobenzoate **(R)-15** (0.3 g, 0.81 mol), di-*tert*-butyl [(4-bromo-5-cyano-1-methyl-1H-pyrazol-3-yl)methyl]imidodicarbonate **16** (505 mg, 1.22 mmol) and B₂Pin₂ (620 g, 2.44 mmol) in methanol (60 ml) was added cataCXium A (40 mg, 0.1057 mmol) and Pd(OAc)₂ (25 mg, 0.1057 mmol). The reaction mixture was degassed three times with N₂ gas and an aqueous solution of NaOH (12 ml, 0.14 M) was added to the mixture under a blanket of N₂ gas. The resulting mixture was again degassed with N₂ gas three times prior to being heated to reflux for 16 h and the reaction was monitored by TLC (petroleum ether/EtOAc 3:1). Upon completion of the reaction, the resulting mixture was extracted with EtOAc (3 × 50 ml), and the combined organic layers washed with brine (2 × 30 ml), dried over Na₂SO₄ and concentrated *in vacuo* to afford a residue, which was purified via column chromatography on silica gel eluting with petroleum ether/EtOAc (from 10:1 to 5:1) to give methyl 2-[(1R)-1-{[2-amino-5-(3-{[bis(tert-butoxycarbonyl)amino]methyl}-5-cyano-1-methyl-1H-pyrazol-4-yl)pyridine-3-yl]oxy}ethyl]-4-fluorobenzoate **(R)-17** (340 mg, 52%) as a brown solid. LCMS (APCI) *m/z* 647.1 [M + Na]⁺.

**Methyl 2-[(1R)-1-({2-amino-5-[3-(aminomethyl)-5-cyano-1-methyl-1H-pyr-azol-4-yl]pyridin-3-yl}oxy)ethyl]-4-fluorobenzoate (R)-18.** To a stirred solution of **(R)-17** (340 mg, 0.42 mmol) in CH₂Cl₂ (2 ml) was added in a dropwise manner at room temperature hydrogen chloride (4 M in dioxane, 10 ml). After the addition was complete, the reaction was stirred at room temperature for 2 h after which time LCMS showed that the reaction was complete. The reaction was concentrated *in vacuo* to afford crude methyl 2-[(1R)-1-({2-amino-5-[3-(aminomethyl)-5-cyano-1-methyl-1H-pyrazol-4-yl]pyridin-3-yl}oxy)ethyl]-4-fluorobenzoate **(R)-18,** which was used in the next step without any further purification. LCMS (APCI) *m/z* 425.2 [M + H]⁺.

**2-[(1R)-1-({2-amino-5-[3-(aminomethyl)-5-cyano-1-methyl-1H-pyrazol-4-yl]pyridin-3-yl}oxy)ethyl]-4-fluorobenzoic acid (R)-19.** A mixture **(R)-18** (~300 mg) and KOH (0.395 g, 7.0 mmol) in methanol (15 ml) was stirred at 50 °C for 36 h after which time LCMS showed that the reaction was complete. The solvent was removed *in vacuo* to afford a residue, which was treated with 1 N aq. HCl to adjust the pH to ~5. The solution was saturated with solid NaCl, and then extracted with EtOAc (5 × 30 ml). The combined EtOAc layers were dried over Na₂SO₄ and concentrated *in vacuo* to give 2-[(1R)-1-({2-amino-5-[3-(aminomethyl)-5-cyano-1-methyl-1H-pyrazol-4-yl]pyridin-3-yl}oxy)ethyl]-4-

fluorobenzoic acid (180 mg) as a brown solid, which was further purified by preparative HPLC to give 2-[(1*R*)-1-({2-amino-5-[3-(aminomethyl)-5-cyano-1-methyl-1*H*-pyrazol-4-yl]pyridin-3-yl}oxy)ethyl]-4-fluorobenzoic acid **(*R*)-19** (70 mg, 41%) as a white solid. LCMS (APCI) *m/z* 411.1 [M + H]$^+$.

### (10*R*)-7-amino-12-fluoro-2,10-dimethyl-15-oxo-10,15,16,17-tetrahydro-2*H*-8,4-(metheno) pyrazolo[4,3-h][2,5,11]benzoxadiazacyclotetradecine-3-carbonitrile (*R*)-2.

To a solution of **(*R*)-19** (70 mg, 0.17 mmol) and *N,N*-Diisopropylethylamine (DIPEA) (33 mg, 0.256 mmol) in DMF (25 ml) was added at −35 °C a solution of 1-Hydroxybenzotriazole (HOBT) (35 mg, 0.256 mmol) and 1-Ethyl-3-(3-dimethylaminopropyl)carbodiimide (EDCI) (33 mg, 0.256 mmol) in DMF (10 ml). After the addition was completed, the resulting mixture was stirred at 80 °C for 72 h, after which time LCMS showed the reaction to be complete. The mixture was poured into ice water (50 ml) and extracted with EtOAc (5 × 40 ml). The combined EtOAc layers were washed with brine (5 × 20 ml), dried over Na$_2$SO$_4$ and concentrated *in vacuo* to give a residue, which was purified initially via preparative TLC, and then subsequently by preparative HPLC to give (10*R*)-7-amino-12-fluoro-2,10-dimethyl-15-oxo-10,15,16,17-tetrahydro-2*H*-8,4-(metheno)-pyrazolo[4,3-h][2,5,11]benzoxadiazacyclotetradecine-3-carbonitrile **(*R*)-2** (11.5 mg, 17%) as a white solid, which was shown by both NMR and HPLC to be a mixture of *cis* and *trans* diastereomers (1:3). LCMS (APCI) *m/z* 392.2 [M + H]$^+$; $^1$H NMR Major (600 MHz, CD$_3$OD) δ 7.64 (dd, *J* = 10.8, 2.6 Hz, 1H), 7.62 (s, 1 H), 7.29 (dd, *J* = 8.5, 5.5 Hz, 7.25 (s, 1 H), 7.03 (td, *J* = 8.3, 2.6 Hz, 1 H), 6.28 (q, *J* = 7.5 Hz, 1 H), 5.59 (d, *J* = 16.8 Hz, 1H), 4.36 (d, *J* = 16.8 Hz, 1 H), 4.02 (s, 3 H), 1.81 (d, *J* = 6.4 Hz, 3 H); $^{13}$C NMR (151 MHz, CD$_3$OD) δ 172.23, 165.12 (d, *J* = 248.3 Hz), 153.30, 145.84, 143.83 (d, *J* = 7.2 Hz), 139.99, 139.58, 134.88 (d, *J* = 3.2 Hz), 129.06 (d, *J* = 8.5 Hz), 125.34, 125.25, 116.48 (d, *J* = 22.1 Hz), 115.71 (d, *J* = 22.6 Hz), 115.22, 114.83, 111.98, 73.43, 39.32, 38.79, 20.89; $^{19}$F NMR (377 MHz, CD$_3$OD) δ −113.68, −111.48 (note that it is negative relative to the reference CFCl$_3$ at 0 p.p.m. for $^{19}$F). Minor (600 MHz, CD$_3$OD) δ 7.62 (s, 1H), 7.52 (dd, *J* = 9.9, 2.6 Hz, 1 H), 7.43 (dd, *J* = 8.5, 5.4 Hz, 1 H), 7.14 (td, *J* = 8.3, 2.6 Hz, 1 H), 5.78 (q, *J* = 6.6 Hz, 1 H), 4.47 (d, *J* = 14.5 Hz, 1 H), 4.03 (s, 3 H), 3.97 (d, *J* = 14.5 Hz, 1 H), 1.79 (s, 3 H); $^{13}$C NMR (151 MHz, CD$_3$OD) δ 172.76, 165.45 (d, *J* = 249.7 Hz), 151.94, 146.69, 144.53 (d, *J* = 7.3 Hz), 140.47, 137.10, 132.70 (d, *J* = 2.8 Hz), 130.00 (d, *J* = 8.9 Hz), 127.57, 119.79, 116.57 (d, *J* = 22.3 Hz), 115.51 (d, *J* = 22.7 Hz), 115.34, 113.45, 111.65, 72.55, 42.15, 38.89, 22.57; $^{19}$F NMR (377 MHz, CD$_3$OD) δ −113.68, −110.59 (note that it is negative relative to the reference CFCl$_3$ at 0 p.p.m. for $^{19}$F).

### Methyl 2-[(1*R*)-1-({2-[bis(tert-butoxycarbonyl)amino]-5-bromopyridin-3-yl}oxy)ethyl]-4-fluorobenzoate (*R*)-20.

To a stirred yellow solution of methyl 2-{(1*R*)-1-[(2-amino-5-bromopyridin-3-yl)oxy]ethyl}-4-fluorobenzoate **(*R*)-15** (3.0 g, 8.126 mmol) in anhydrous CH$_2$Cl$_2$ (40 ml) was added DIPEA (3.15 g, 24.4 mmol) and DMAP (496 mg, 4.06 mmol). To this mixture was added in a dropwise manner, a solution of Boc$_2$O (14.2 g, 65.0 mmol) in CH$_2$Cl$_2$ (10 ml). The yellow reaction mixture was stirred at 10 °C for 20 h. TLC (petroleum ether/EtOAc 6/1, ultraviolet, 254 nm) showed that the starting material had been consumed, and a new main product was detected. The reaction mixture was diluted with CH$_2$Cl$_2$ (100 ml), washed with brine (20 ml), dried over Na$_2$SO$_4$, filtered and concentrated to give a residual yellow oil, which was purified by silica gel chromatography (petroleum ether/EtOAc 6/1) to give methyl 2-[(1*R*)-1-({2-[bis(*tert*-butoxycarbonyl)amino]-5-bromopyridin-3-yl}oxy)ethyl]-4-fluorobenzoate **(*R*)-20** (3.5 g, 76%) as a white solid. $^1$H NMR (400 MHz, CDCl$_3$) δ 8.09–8.07 (m, 2 H), 7.32 (d, *J* = 2 Hz, 1 H), 7.15 (s, 1 H), 7.09–7.04 (m, 1 H), 6.42 (q, *J* = 6.4 Hz, 1 H), 3.97 (s, 3 H), 1.58 (d, *J* = 6.0 Hz, 3 H), 1.45 (s, 18 H).

### 2-[(1*R*)-1-({2-[bis(tert-butoxycarbonyl)amino]-5-bromopyridin-3-yl}oxy)ethyl]-4-fluorobenzoic acid (*R*)-21.

To a suspension of methyl 2-[(1*R*)-1-({2-[bis(*tert*-butoxycarbonyl)amino]-5-bromopyridin-3-yl}oxy)ethyl]-4-fluorobenzoate **(*R*)-20** (3.5 g, 6.147 mmol) in water (17.5 ml) was added tetrahydrofuran (THF) (15.4 g, 214 mmol) followed by an aqueous solution of NaOH (9.22 ml, 9.22 mmol, 1 M). The cloudy mixture was stirred at 30 °C for 48 h. TLC (petroleum ether/EtOAc 6/1) indicated that the starting material had been consumed. The mixture was acidified with 1 N HCl to pH ∼ 2, and the resulting colourless solution was concentrated to remove the THF. The suspension was extracted with EtOAc (3 × 250 ml), and the combined organic layers washed with brine (100 ml), dried over Na$_2$SO$_4$, filtered and concentrated. The residue was diluted with petroleum ether/EtOAc (10: 1, 600 ml), and the suspension filtered and dried to give 2-[(1*R*)-1-({2-[bis(*tert*-butoxycarbonyl)amino]-5-bromopyridin-3-yl}oxy)ethyl]-4-fluorobenzoic acid **(*R*)-21** (3.2 g, 94%) as a white solid. $^1$H NMR (400 MHz, CDCl$_3$) δ 8.24–8.20 (m, 1 H), 8.13 (s, 1 H), 7.37 (d, *J* = 2.4 Hz, 1 H), 7.15 (s, 1 H), 7.15–7.09 (m, 1 H), 6.47–6.45 (m, 1 H), 1.65 (d, *J* = 6.4 Hz, 3 H), 1.47 (s, 18 H).

### 3-(Aminomethyl)-1-methyl-1*H*-pyrazole-5-carbonitrile hydrochloride 22.

To a stirred colourless solution of di-*tert*-butyl [(5-cyano-1-methyl-1*H*-pyrazol-3-yl)methyl]imidodicarbonate **10** (2.0 g, 5.95 mmol) in CH$_2$Cl$_2$ (10 ml) was added HCl(g) in dioxane (30 ml, 4 M) at 0 °C. After the addition was completed, the solution was allowed to warm slowly to 15 °C, and stirred at this temperature for

2 h. During the stirring, a white solid precipitated, and TLC (petroleum ether/EtOAc 10/1) showed that the di-*tert*-butyl [(5-cyano-1-methyl-1*H*-pyrazol-3-yl)methyl]imidodicarbonate had been consumed. The white suspension was concentrated to give 3-(aminomethyl)-1-methyl-1H-pyrazole-5-carbonitrile hydrochloride **22** (1 g, 97%) as a white solid, which was used without further purification.

### Tert-Butyl N-{5-bromo-3-[(1*R*)-1-(2-{[(5-cyano-1-methyl-1H-pyrazol-3-yl)methyl]carbamoyl}-5-fluorophenyl)ethoxy]pyridin-2-yl}-N-(tert-butoxycarbonyl) glycinate (*R*)-23.

To a white suspension of 2-[(1*R*)-1-({2-[bis(*tert*-butoxycarbonyl)amino]-5-bromopyridin-3-yl}oxy)ethyl]-4-fluorobenzoic acid **(*R*)-21** (3.4 g, 6.12 mmol) and 3-(aminomethyl)-1-methyl-1*H*-pyrazole-5-carbonitrile hydrochloride **22** (1.37 g, 7.96 mmol) in 2-MeTHF (40 ml) was added at <35 °C under a N$_2$ atmosphere with Et$_3$N (2.48 g, 24.5 mmol). After the addition was completed, the mixture was stirred at 10 ∼ 15 °C for 30 min. T3P (7.79 g, 12.2 mmol, 50% in EtOAc) was added in a dropwise manner at 10–20 °C, and then the white cloudy reaction was stirred at the same temperature for 17 h. TLC (petroleum ether/EtOAc 1/1, $R_f$ ∼ 0.6) showed that **(*R*)-21** had been consumed. The light-yellow and cloudy reaction mixture was diluted with EtOAc (100 ml) and nH$_2$O (40 ml), and the layers were separated. The organic layer was washed with saturated NaHCO$_3$ (25 ml), brine (25 ml), dried over Na$_2$SO$_4$, filtered and concentrated to give the crude product as a solid, which was purified by silica gel chromatography with petroleum ether/EtOAc 2:1 to give *tert*-butyl N-{5-bromo-3-[(1*R*)-1-(2-{[(5-cyano-1-methyl-1*H*-pyrazol-3-yl)methyl]carbamoyl}-5-fluorophenyl)ethoxy]pyridin-2-yl}-*N*-(*tert*-butoxycarbonyl)glycinate **(*R*)-23** (3.4 g, 83%) as a white solid. LCMS (APCI) *m/z* 696.8 [M + Na]$^+$; $^1$H NMR (400 MHz, CD$_3$OD) δ 8.07–8.02 (m, 1 H), 7.50–7.46 (m, 2 H), 7.25–7.20 (m, 2 H), 7.03–6.99 (m, 1 H), 6.78–6.77 (m, 1 H), 6.60–6.59 (m, 1 H), 6.09–6.07 (m, 1 H), 4.65 (t, *J* = 5.6 Hz, 2 H), 4.01 (s, 3 H), 1.60 (d, *J* = 6.4 Hz, 3 H), 1.42 (s, 18 H).

### Di-tert-butyl [(10*R*)-3-cyano-12-fluoro-2,10-dimethyl-15-oxo-10,15,16,17-tetrahydro-2*H*-8,4-(metheno)pyrazolo[4,3-h][2,5,11]benzoxadiazacyclotetradecin-7-yl]imidodicarbonate (*R*)-3.

Argon was bubbled through a colourless solution of *tert*-butyl N-{5-bromo-3-[(1*R*)-1-(2-{[(5-cyano-1-methyl-1*H*-pyrazol-3-yl)methyl]carbamoyl}-5-fluorophenyl)ethoxy]pyridin-2-yl}-*N*-(*tert*-butoxycarbonyl)glycinate **(*R*)-23** (250 mg, 0.37 mmol), KOAc (120 mg, 1.22 mmol), Pd(OAc)$_2$ (6.33 mg, 0.028 mmol) and H$_2$O (6.69 mg, 0.37 mmol) in 2-methyl-2-butanol (10 ml) for 3 min. CataCXiumA (10.1 mg, 0.028 mmol) was added, and the mixture was bubbled with Ar for a further 3 min. The yellow reaction mixture was stirred at 140 °C under microwave heating for 1 h. TLC (petroleum ether/EtOAc 1:1) showed that *tert*-butyl N-{5-bromo-3-[(1*R*)-1-(2-{[(5-cyano-1-methyl-1*H*-pyrazol-3-yl)methyl]carbamoyl}-5-fluorophenyl)ethoxy]pyridin-2-yl}-*N*-(*tert*-butoxycarbonyl)glycinate had been consumed, and two new spots were detected. The yellow mixture was poured into ice-H$_2$O (10 ml), and extracted with EtOAc (3 × 25 ml). The combined organic layers were washed with brine (20 ml), dried over Na$_2$SO$_4$, filtered and concentrated to give the crude product as a brown solid. Initial purification was achieved by preparative TLC (petroleum ether/EtOAc 1/3, $R_f$ ∼ 0.5) to afford the desired compound as a white solid, which was further purified by preparative HPLC to afford di-*tert*-butyl [(10*R*)-3-cyano-12-fluoro-2,10-dimethyl-15-oxo-10,15,16,17-tetrahydro-2*H*-8,4-(metheno)pyrazolo[4,3-h][2,5,11]benzoxadiazacyclotetradecin-7-yl]imidodicarbonate **(*R*)-3** (99 mg, 45%) as an off-white solid as a mixture of *cis* and *trans* diastereomers. LCMS (APCI) *m/z* 593.1 [M + H]$^+$; mixture of *cis* and *trans* diastereomers $^1$H NMR (400 MHz, CD$_3$OD) δ 8.18 (s, 1 H), 7.71 (s, 0.35 H), 7.55–7.21 (m, 2 H), 7.18–7.03 (m, 1 H), 6.51–6.49 (m, 0.65 H), 5.83–5.82 (m, 0.35 H), 5.72–5.68 (m, 0.7 H), 4.62–4.58 (m, 1 H), 4.43–4.41 (m, 3 H), 1.82–1.80 (m, 3 H), 1.48–1.41 (m, 18 H); $^{13}$C NMR (101 MHz, CD$_3$OD) δ 171.02, 170.74, 165.04, 162.57, 150.94, 148.00, 145.46, 142.58, 138.93. 136.81, 133.03, 127.87, 126.58, 126.44, 122.06, 114.53, 115.31, 114.74, 114.61, 114.52, 110.13, 83.29, 83.15, 72.06, 40.76, 37.91, 37.63, 26.68, 21.33, 19.58; $^{19}$F NMR (377 MHz, CD$_3$OD) δ −109.99, −109.83 (note that it is negative relative to the reference CFCl$_3$ at 0 p.p.m. for $^{19}$F).

The synthesis of methyl 2-{1-[(2-amino-5-bromopyridin-3-yl)oxy]ethyl}-4-nitrobenzoate **31** (Supplementary Fig. 4) is described below.

### 1-(2-Amino-5-nitrophenyl)ethanone 25.

A mixture of 1-(2-fluoro-5-nitrophenyl) ethanone **24** (50 g, 273 mmol) in 28% NH$_4$OH (500 ml) and THF (500 ml) was stirred at 70 °C for 16 h. TLC (petroleum ether/EtOAc = 5:1) showed the majority of 1-(2-fluoro-5-nitrophenyl)ethanone had been consumed with one new spot detected. The mixture was concentrated under vacuum, and the aqueous solution was extracted with EtOAc (3 × 400 ml). The organic layers were washed with brine (400 ml), dried over Na$_2$SO$_4$ and concentrated in vacuum to afford a residue, which was purified by silica gel chromatography (petroleum ether/EtOAc from 10:1 to 3:1) to give 1-(2-amino-5-nitrophenyl)ethanone **25** (42 g, 85%) as a yellow solid. $^1$H NMR (400 MHz, CDCl$_3$) δ 8.72 (d, *J* = 2 Hz, 1 H), 8.15–8.12 (m, 1 H), 6.66 (d, *J* = 8.8 Hz, 1 H), 2.67 (s, 3 H).

### 1-(2-Iodo-5-nitrophenyl)ethanone 26.

To a stirred solution of 1-(2-amino-5-nitrophenyl)ethanone **25** (42 g, 233 mmol) and isoamyl nitrite (54.6 g, 466 mmol)

in CH$_3$CN (420 ml) was added CH$_2$I$_2$ (125 g, 466 mmol) at 15 °C in a single portion. The mixture was stirred at 80 °C for 4 h. TLC (petroleum ether/EtOAc = 5/1) indicated that 1-(2-amino-5-nitrophenyl)ethanone had been consumed with several new spots being detected. The mixture was concentrated under vacuum, and the residue purified by silica gel chromatography (petroleum ether/EtOAc = 10:1) to give 1-(2-iodo-5-nitrophenyl)ethanone **26** (36 g, 53%) as a yellow solid. $^1$H NMR (400 MHz, CDCl$_3$) δ 8.27 (d, $J$ = 2.4 Hz, 1 H), 8.16 (d, $J$ = 8.4 Hz, 7.97–7.94 (m, 1 H), 2.69 (s, 3 H).

**1-(2-Iodo-5-nitrophenyl)ethanol 27.** To a stirred solution of 1-(2-iodo-5-nitrophenyl)ethanone **26** (36 g, 123.69 mmol) in EtOH (360 ml) was added NaBH$_4$ (5.62 g, 148 mmol) in portions at 0 °C. After the addition, the reaction mixture was stirred at room temperature for 2 h. TLC (petroleum ether/EtOAc = 5:1) indicated that the 1-(2-iodo-5-nitrophenyl)ethanone had been consumed with one new spot being detected. The reaction mixture was quenched with water (20 ml), and then concentrated. The residue was diluted with EtOAc (400 ml) and water (200 ml). The organic layer was washed with brine (200 ml), dried over Na$_2$SO$_4$ and concentrated in vacuo to afford a residue, which was purified by silica gel chromatography (petroleum ether/EtOAc from 10:1 to 5:1) to give 1-(2-iodo-5-nitrophenyl)ethanol **27** (30 g, 83%) as a yellow solid. $^1$H NMR (400 MHz, CDCl$_3$) δ 8.43–8.42 (m, 1 H), 8.01–7.98 (m, 1 H), 7.81 (d, $J$ = 8.4 Hz, 1 H), 5.14–5.10 (m, 1 H), 2.15–2.14 (m, 1 H), 1.52–1.49 (m, 3 H).

**1-(2-Iodo-5-nitrophenyl)ethyl methanesulfonate 28.** To a stirred solution of 1-(2-iodo-5-nitrophenyl)ethanol **27** (30 g, 102.4 mmol), DMAP (6.25 g, 51.2 mmol) and TEA (31.1 g, 307 mmol) in CH$_2$Cl$_2$ (300 ml) was added in a dropwise manner MsCl (23.5 g, 205 mmol) at 0 °C under a N$_2$ atmosphere. Then, the mixture was allowed to warm to room temperature and stirred for 16 h. TLC (petroleum ether/EtOAc = 5:1) showed that 1-(2-iodo-5-nitrophenyl)ethanol had been consumed, one new spot being detected. The mixture was diluted with CH$_2$Cl$_2$ (100 ml), washed with water (100 ml) and brine (3 × 100 ml), dried over Na$_2$SO$_4$ and concentrated to dryness to afford 1-(2-iodo-5-nitrophenyl)ethyl methanesulfonate **28** (37.99 g), which was used directly in the next step without further purification.

**3-[1-(2-Iodo-5-nitrophenyl)ethoxy]pyridin-2-amine 29.** A mixture of 1-(2-iodo-5-nitrophenyl)ethyl methanesulfonate **28** (37.99 g, 102.37 mmol), 3-hydroxy-2-aminopyridine (12.4 g, 113 mmol) and Cs$_2$CO$_3$ (100 g, 307 mmol) in acetone (400 ml) was stirred at 45 °C for 16 h. TLC (petroleum ether/EtOAc) showed that 1-(2-iodo-5-nitrophenyl)ethyl methanesulfonate had been consumed with several new spots detected. The mixture was filtered, and the filtrate was concentrated under vacuum. The residue was dissolved in EtOAc (500 ml), washed with water (200 ml) and brine (200 ml), dried over Na$_2$SO$_4$ and concentrated in vacuo. The residue was purified by silica gel chromatography (petroleum ether/EtOAc from 5:1 to 1:1) to give 3-[1-(2-iodo-5-nitrophenyl)ethoxy]pyridin-2-amine **29** (15.1 g, 38%) as a yellow solid. $^1$H NMR (400 MHz, CDCl$_3$) δ 8.20 (d, $J$ = 2.8 Hz, 1 H), 8.05 (d, $J$ = 8.4 Hz, 1 H), 7.85–7.83 (m, 1 H), 7.64 (d, $J$ = 5.2 Hz, 1 H), 6.52–6.45 (m, 2 H), 5.48 (q, $J$ = 6.4 Hz, 1 H), 4.80 (br s, 1 H), 1.67 (d, $J$ = 6.4 Hz, 3 H).

**Methyl 2-{1-[(2-aminopyridin-3-yl)oxy]ethyl}-4-nitrobenzoate 30.** A mixture of 3-[1-(2-iodo-5-nitrophenyl)ethoxy]pyridin-2-amine **29** (15.1 g, 39.21 mmol), TEA (7.93 g, 78.4 mmol) and (PPh$_3$)$_2$PdCl$_2$ (1.93 g, 2.74 mmol) in MeOH (180 ml) and CH$_3$CN (300 ml) was stirred at 50 °C under 4 bar of CO pressure for 16 h. TLC (petroleum ether/EtOAc = 1:1) showed that 3-[1-(2-iodo-5-nitrophenyl)ethoxy]pyridin-2-amine had been consumed, with several new spots being detected. The mixture was concentrated in vacuo, and the residue dissolved in EtOAc (500 ml) and washed with water (200 ml) and brine (200 ml). The organic phase was dried over Na$_2$SO$_4$, filtered and concentrated under vacuum, and the residue purified by silica gel chromatography (petroleum ether/EtOAc from 3:1 to 1:1) to give methyl 2-{1-[(2-aminopyridin-3-yl)oxy]ethyl}-4-nitrobenzoate **30** (7.51 g, 60%) as a yellow solid. $^1$H NMR (400 MHz, CDCl$_3$) δ 8.41 (d, $J$ = 2.0 Hz, 1 H), 8.19–8.17 (m, 2 H), 7.76 (d, $J$ = 5.2 Hz, 1 H), 6.70 (d, $J$ = 6.8 Hz, 1 H), 6.47–6.45 (m, 1 H), 6.33–6.31 (m, 1 H), 5.50 (br s, 2 H), 4.01 (s, 3 H), 1.71 (d, $J$ = 6.0 Hz, 3 H).

**Ethyl 2-{1-[(2-amino-5-bromopyridin-3-yl)oxy]ethyl}-4-nitrobenzoate 31.** To a stirred solution of methyl 2-{1-[(2-aminopyridin-3-yl)oxy]ethyl}-4-nitrobenzoate **30** (7.51 g, 23.67 mmol) in CH$_3$CN (100 ml) was added N-Bromosuccinimide (NBS) (4.21 g, 23.7 mmol) in portions at 0 °C. After the addition was complete, the resulting mixture was stirred at 0 °C for 2 h. TLC (petroleum ether/EtOAc = 1:1) showed that methyl 2-{1-[(2-aminopyridin-3-yl)oxy]ethyl}-4-nitrobenzoate had been consumed, with several new spots being detected. The reaction mixture was quenched with saturated NaHCO$_3$ (50 ml) and extracted with EtOAc (2 × 300 ml). The organic layer was washed with brine (2 × 100 ml), dried over Na$_2$SO$_4$ and concentrated in vacuo. The residue was purified by silica gel chromatography (petroleum ether/EtOAc from 10:1 to 2:1) to give methyl 2-{1-[(2-amino-5-bromopyridin-3-yl)oxy]ethyl}-4-nitrobenzoate **31** (3.76 g, 40%) as a red solid. LCMS (APCI) $m/z$ 397.8 [M + H]$^+$; $^1$H NMR (400 MHz, CDCl$_3$) δ 8.43

(d, $J$ = 1.6 Hz, 1 H), 8.22–8.13 (m, 2 H), 7.69 (s, 1 H), 6.80 (d, $J$ = 1.6 Hz, 1 H), 6.30–6.28 (m, 1 H), 4.82 (br s, 2 H), 4.03 (s, 3 H), 1.70 (d, $J$ = 6.4 Hz, 3 H).

The synthesis of the nitro-bearing head group is shown in Supplementary Fig. 4, and for a large part mirrors the chemistry utilized for the formation of the head group of lorlatinib. Starting with compound **31** (Supplementary Fig. 5), the macrocycle **4** can be accessed through a series of well-established transformations, as shown in Supplementary Fig. 4. Suzuki coupling with pyrazole **32** leads to **33**, which undergoes controlled ester hydrolysis, thus minimizing the potential for side reactions such as nitrile hydrolysis, followed by Boc deprotection to provide the macrolactamization precursor. Addition of this precursor to a solution of HATU in DMF leads to **5**, which could be smoothly protected to provide **4**.

The synthesis of 7-amino-2,10,16-trimethyl-12-nitro-15-oxo-10,15,16,17-tetrahydro-2H-8,4-(metheno)pyrazolo [4,3-h][2,5,11]benzoxadiazacyclotetradecine-3-carbonitrile **5** and Di-tert-butyl [3-cyano-2,10,16-trimethyl-12-nitro-15-oxo-10,15,16,17-tetrahydro-2H-8,4-(metheno) pyrazolo[4,3-h][2,5,11] benzoxadiazacyclotetradecin-7-yl]imidodicarbonate **4** (Supplementary Fig. 5) is described below.

**Methyl 2-(1-{[2-amino-5-(3-{[(tert-butoxycarbonyl)(methyl)amino]methyl}-5-cyano-1-methyl-1H-pyrazol-4-yl)pyridin-3-yl]oxy}ethyl)-4-nitrobenzoate 33.** A mixture of methyl 2-{1-[(2-amino-5-bromopyridin-3-yl)oxy]ethyl}-4-nitrobenzoate **31** (2.5 g, 6.31 mmol), tert-butyl [[(4-bromo-5-cyano-1-methyl-1H-pyrazol-3-yl)methyl]methylcarbamate **32** (3.12 g, 9.47 mmol), B$_2$Pin$_2$ (2.08 g, 8.20 mmol) and CsF (4.79 g, 31.6 mmol) in MeOH (50 ml) was warmed to 50 °C under a N$_2$ atmosphere. Pd(OAc)$_2$ (0.142 g, 0.631 mmol) and cataCXium A (0.452 g, 1.26 mmol) in toluene (50 ml) were added to the mixture, which was then stirred at reflux under N$_2$ for 2 h. TLC (petroleum ether/EtOAc = 2:1) showed that methyl 2-{1-[(2-amino-5-bromopyridin-3-yl)oxy]ethyl}-4-nitrobenzoate had been consumed with several new spots being detected. The reaction mixture was cooled, diluted with EtOAc (200 ml) and then filtered. The filtrate was concentrated in vacuo to give crude product, which was purified by silica gel chromatography (petroleum ether/EtOAc from 5:1 to 1:1) to give methyl 2-(1-{[2-amino-5-(3-{[(tert-butoxycarbonyl)(methyl)amino]methyl}-5-cyano-1-methyl-1H-pyrazol-4-yl)pyridin-3-yl]oxy}ethyl)-4-nitrobenzoate **33** (1.83 g, 51%) as a yellow solid. LCMS (APCI) $m/z$ 566.1 [M + H]$^+$.

**2-(1-{[2-Amino-5-(3-{[(tert-butoxycarbonyl)(methyl)amino]methyl}-5-cyano-1-methyl-1H-pyrazol-4-yl)pyridin-3-yl]oxy}ethyl)-4-nitrobenzoic acid 34.** A mixture of methyl 2-(1-{[2-amino-5-(3-{[(tert-butoxycarbonyl)(methyl)amino]methyl}-5-cyano-1-methyl-1H-pyrazol-4-yl)pyridin-3-yl]oxy}ethyl)-4-nitrobenzoate **33** (1.0 g, 1.77 mmol) and KOH (992 mg, 17.7 mmol) in MeOH (50 ml) and water (1 ml) was stirred at 15 °C for 16 h. TLC (petroleum ether/EtOAc = 1:1) showed that **33** had been consumed with one new spot being detected. The reaction mixture was neutralized to pH = 7 with 5% aqueous HCl solution. The mixture was concentrated under vacuum, and the residue acidified with 5% HCl aqueous solution to pH = 3. The mixture was extracted with EtOAc (3 × 30 ml, and the combined organic layers washed with brine (30 ml), dried over Na$_2$SO$_4$ and concentrated in vacuo to give 2-(1-{[2-amino-5-(3-{[(tert-butoxycarbonyl)(methyl)amino]methyl}-5-cyano-1-methyl-1H-pyrazol-4-yl)pyridin-3-yl]oxy}ethyl)-4-nitrobenzoic acid **34** (975 mg) as a yellow solid, which was used directly in the next step. LCMS (APCI) $m/z$ 552.1 [M + H]$^+$.

**2-{1-[(2-Amino-5-{5-cyano-1-methyl-3-[(methylamino)methyl]-1H-pyrazol-4-yl}pyridin-3-yl)oxy]ethyl}-4-nitrobenzoic acid 35.** To a stirred solution of 2-(1-{[2-amino-5-(3-{[(tert-butoxycarbonyl)(methyl)amino]methyl}-5-cyano-1-methyl-1H-pyrazol-4-yl)pyridin-3-yl]oxy} ethyl)-4-nitrobenzoic acid **34** (1.95 g, 3.54 mmol) in CH$_2$Cl$_2$ (20 ml) was added in a dropwise manner ∼4 N HCl (g)/EtOAc (20 ml) at 0 °C. After addition was completed, the resulting mixture was stirred at 20 °C for 4 h. LCMS indicated that the reaction was completed and the main peak was the desired product. The reaction mixture was concentrated in vacuo to afford the hydrochloride salt of 2-{1-[(2-amino-5-{5-cyano-1-methyl-3-[(methylamino)methyl]-1H-pyrazol-4-yl}pyridin-3-yl)oxy]ethyl}-4-nitrobenzoic acid **35** (∼3.54 mmol) as a yellow solid, which was used directly in the next step without further purification. LCMS (APCI) $m/z$ 451.9 [M + H]$^+$.

**7-Amino-2,10,16-trimethyl-12-nitro-15-oxo-10,15,16,17-tetrahydro-2H-8,4-(metheno)pyrazolo [4,3-h][2,5,11]benzoxadiazacyclotetradecine-3-carbonitrile 5.** A mixture of the hydrochloride salt of 2-{1-[(2-amino-5-{5-cyano-1-methyl-3-[(methylamino)methyl]-1H-pyrazol-4-yl}pyridin-3-yl)oxy]ethyl}-4-nitrobenzoic acid **35** (200 mg, 0.41 mmol) and DIPEA (530 mg, 4.1 mmol) in DMF (20 ml) was added dropwise to a mixture of HATU (234 mg, 0.615 mmol) in DMF (20 ml) at 0 °C. After addition, the mixture was stirred at 0 °C for 1 h. LCMS indicated that ∼30% of the desired product was detected. The solvent was evaporated under vacuum, and water (60 ml) was added to the mixture, which was extracted with EtOAc (3 × 100 ml). The organic layer was washed with aqueous Na$_2$CO$_3$ aqueous solution (5 × 50 ml), brine (5 × 50 ml), dried over Na$_2$SO$_4$ and concentrated in vacuum to give the crude product, which was purified by silica gel chromatography (silica gel, eluent: CH$_2$Cl$_2$/methanol = 10:1, $R_f$ ∼0.4) to

give the crude compound (70 mg), which was further purified by preparative TLC (silica gel, $CH_2Cl_2$/MeOH = 10:1, $R_f \sim 0.4$) to give 7-amino-2,10,16-trimethyl-12-nitro-15-oxo-10,15,16,17-tetrahydro-2$H$-8,4-(metheno)pyrazolo[4,3-$h$][2,5,11]benzoxadiazacyclotetradecine-3-carbonitrile 5 (45 mg, 25%) as a yellow solid. LCMS (APCI) $m/z$ 434.0 $[M + H]^+$; $^1$H NMR (400 MHz, CDCl$_3$) δ 8.48 (d, $J = 1.2$ Hz, 1 H), 8.18 (d, $J = 2.4$ Hz, 1 H), 7.84 (s, 1 H), 7.42 (d, $J = 8.4$ Hz, 1 H), 6.86 (s, 1 H), 5.77 (q, $J = 6.4$ Hz, 1 H), 4.94 (bs s, 2 H), 4.39 (q, $J = 14.0$ Hz, 2 H), 4.08 (s, 3 H), 3.17 (s, 3 H), 1.85 (d, $J = 6.4$ Hz, 3 H).

**Di-tert-butyl [3-cyano-2,10,16-trimethyl-12-nitro-15-oxo-10,15,16,17-tetrahydro-2H-8,4-(metheno) pyrazolo[4,3-h][2,5,11]benzoxadiazacyclote-tradecin-7-yl]imidodicarbonate 4.** A mixture of 7-amino-2,10,16-trimethyl-12-nitro-15-oxo-10,15,16,17-tetrahydro-2$H$-8,4-(metheno)pyrazolo[4,3-$h$][2,5,11]benzoxadiazacyclotetradecine-3-carbonitrile 5 (300 mg, 0.692 mmol), (Boc)$_2$O (1,510 mg, 6.92 mmol) and DMAP (42.3 mg, 0.346 mmol) in $CH_2Cl_2$ (20 ml) was stirred at 15 °C for 16 h. TLC ($CH_2Cl_2$/MeOH = 20:1) showed that the 7-amino-2,10,16-trimethyl-12-nitro-15-oxo-10,15,16,17-tetrahydro-2$H$-8,4-(metheno) pyrazolo[4,3-$h$][2,5,11]benzoxadiaza cyclotetradecine-3-carbonitrile had been consumed with one new spot being detected. The reaction was concentrated *in vacuo* to give the crude product, which was purified by silica gel chromatography ($CH_2Cl_2$/MeOH = 20:1) to give the crude product. This was further purified by preparative TLC ($CH_2Cl_2$/MeOH = 20:1, $R_f \sim 0.6$, carried out twice) to give pure di-*tert*-butyl [3-cyano-2,10,16-trimethyl-12-nitro-15-oxo-10,15,16,17-tetrahydro-2$H$-8,4-(metheno) pyrazolo [4,3-$h$][2,5,11]benzoxadiazacyclo tetradecin-7-yl] imidodicarbonate 4 (300 mg, 68%) as a white solid. LCMS (APCI) $m/z$ 634.1 $[M + H]^+$; $^1$H NMR (400 MHz, CDCl$_3$) δ 8.50 (d, $J = 2.4$ Hz, 1 H), 8.26 (s, 1 H), 8.18–8.16 (m, 1 H), 7.40 (d, $J = 8.4$ Hz, 1 H), 7.19 (s, 1 H), 5.81–5.76 (m, 1 H), 4.48–4.38 (m, 2 H), 4.13 (s, 3 H), 3.18 (s, 3 H), 1.83 (d, $J = 6.4$ Hz, 3 H), 1.52 (s, 18 H).

**Separation of (1R)-1-(2-iodo-5-nitrophenyl)ethanol and (1S)-1-(2-iodo-5-nitrophenyl)ethanol.** The analytical chiral separation was performed by SFC on a Chiralpak AD-3 (150 × 4.6 mm I.D., 3 μm particle size), which was eluted with 5–40% isopropanol (IPA) (0.05% diethylamine (DEA)) in CO$_2$ (Supplementary Fig. 6). The flow rate of 2.5 ml min$^{-1}$ gave Rt$_{(Peak\ 1)}$ = 5.32 min and Rt$_{(Peak\ 2)}$ = 5.65 min. The racemic mixture of 27 (20 g) was purified by preparative SFC (Chiralpak AD 300 × 50 mm I.D., 10 μm particle size, which was eluted with 25% IPA with 0.05% NH$_4$OH at a flow rate of 200 ml min$^{-1}$) and gave peak 1 ((R)-27) as a yellow solid (8.9 g, 45%) and peak 2 ((S)-27) as a yellow solid (9.3 g, 47%).

(1R)-1-(2-iodo-5-nitrophenyl)ethanol (Peak 1). 99% ee. Shown to be (R)-enantiomer by VCD calculations.

(1S)-1-(2-iodo-5-nitrophenyl)ethanol (Peak 2). 99% ee. $^1$H NMR (400 MHz, CDCl$_3$) δ 8.42 (d, $J = 5.2$ Hz, 1 H), 7.98 (d, $J = 8.8$ Hz, 1 H), 7.82–7.79 (m, 1 H), 5.12 (q, $J = 6.4$ Hz, 1 H), 2.01 (br s, 1 H), 1.50 (d, $J = 6.4$ Hz, 3 H).

The synthesis of 7-amino-2,10,16-trimethyl-12-nitro-15-oxo-10,15,16,17-tetrahydro-2$H$-8,4-(metheno)pyrazolo[4,3-$h$][2,5,11]benzoxadiazacyclotetradecine-3-carbonitrile, 4 via a direct aryl approach (Supplementary Fig. 7) is described below.

**Methyl 2-[1-({2-[bis(tert-butoxycarbonyl)amino]-5-bromopyridin-3-yl}oxy)ethyl]-4-nitrobenzoate 36.** To a stirred colourless solution of methyl 2-{1-[(2-amino-5-bromopyridin-3-yl)oxy]ethyl}-4-nitrobenzoate 31 (18.9 g, 47.70 mmol) in dry $CH_2Cl_2$ (300 ml) was added DIPEA (18.5 g, 143 mmol) and DMAP (2.91 g, 23.9 mmol). Boc$_2$O (83.3 g, 382 mmol) was then added to the mixture in a dropwise manner, and the resulting brown reaction mixture allowed to stir at 20 °C for 20 h. TLC (petroleum ether/EtOAc 2:1) indicated that most of the methyl 2-{1-[(2-amino-5-bromopyridin-3-yl)oxy]ethyl}-4-nitrobenzoate had been consumed, and a major new spot was detected. The brown solution was diluted with $CH_2Cl_2$ (100 ml), washed with brine (20 ml), dried over Na$_2$SO$_4$, filtered and concentrated to give the residue as a brown oil, which was purified by silica gel chromatography (petroleum ether/EtOAc 3:1) to give methyl 2-[1-({2-[bis(*tert*-butoxycarbonyl)amino]-5-bromopyridin-3-yl}oxy)ethyl]-4-nitrobenzoate 36 (25 g, 88%) as a light-yellow viscous oil.

**2-[1-({2-[Bis(tert-butoxycarbonyl)amino]-5-bromopyridin-3-yl}oxy)ethyl]-4-nitrobenzoic acid 29.** To a yellow suspension of methyl 2-[1-({2-[*bis*(*tert*-butoxycarbonyl)amino]-5-bromopyridin-3-yl}oxy)ethyl]-4-nitrobenzoate 36 (25 g, 41.916 mmol) in water (125 ml) was added at 0 °C, to THF (125 ml), followed by a solution of aqueous NaOH solution (1 M, 62.9 ml, 62.9 mmol). The orange suspension was stirred at 28 °C for 16 h during which time the mixture became clear. TLC (petroleum ether/EtOAc 3:1) indicated that the methyl 2-[1-({2-[*bis*(*tert*-butoxycarbonyl)amino]-5-bromopyridin-3-yl}oxy)ethyl]-4-nitrobenzoate had been consumed. The mixture was acidified with 1 N HCl to pH $\sim$2, and the colourless solution was concentrated to remove the THF, during which time a white solid precipitated. The suspension was extracted with EtOAc (3 × 25 ml). The combined organic layers were washed with brine (25 ml), dried over Na$_2$SO$_4$, filtered and concentrated to give 2-[1-({2-[(*tert*-butoxycarbonyl)amino]-5-bromopyridin-3-yl}oxy)ethyl]-4-nitrobenzoic acid 9 (21.8 g, 89%) as a white solid.

**Di-tert-butyl{5-bromo-3-[1-(2-{[(5-cyano-1-methyl-1H-pyrazol-3-yl) methyl] (methyl) carbamoyl}-5-nitrophenyl) ethoxy]pyridin-2-yl}imidodicarbonate 30.** To a suspension of 2-[1-({2-[bis(*tert*-butoxycarbonyl)amino]-5-bromopyridin-3-yl}oxy)ethyl]-4-nitrobenzoic acid 9 (21.8 g, 37.431 mmol) and 3-(aminomethyl)-1-methyl-1$H$-pyrazole-5-carbonitrile hydrochloride 10 (10.0 g, 44.9 mmol) in 2-MeTHF (220 ml) was added below 35 °C TEA (15.2 g, 150 mmol) under a N$_2$ atmosphere. After the addition was completed, the mixture was stirred at 20 °C for 30 min. T3P (47.6 g, 74.9 mmol, 50% in EtOAc) was added slowly at $\sim$25 °C, and then the brown suspension was allowed to stir at 20 °C for 5 h. TLC (EtOAc) indicated that the 2-[1-({2-[bis(*tert*-butoxycarbonyl)amino]-5-bromopyridin-3-yl}oxy)ethyl]-4-nitrobenzoic acid had been consumed, and a new spot was detected. The brown cloudy solution was diluted with EtOAc (500 ml) and cold H$_2$O (100 ml), and the layers separated. The organic layer was washed with brine (100 ml), dried over Na$_2$SO$_4$, filtered and concentrated to give the crude product, which was purified by silica gel chromatography (petroleum ether/EtOAc 1/1) to give di-*tert*-butyl {5-bromo-3-[1-(2-{[(5-cyano-1-methyl-1$H$-pyrazol-3-yl)methyl](methyl)carbamoyl}-5-nitrophenyl) ethoxy]pyridin-2-yl}imidodicarbonate 11 (21 g, 79% ) as a light-yellow solid.

**Di-tert-butyl [3-cyano-2,10,16-trimethyl-12-nitro-15-oxo-10,15,16,17-tetra-hydro-2H-8,4-(metheno)pyrazolo[4,3-h][2,5,11]benzoxadiazacyclote-tradecin-7-yl]imidodicarbonate 4.** A colourless solution of di-*tert*-butyl {5-bromo-3-[1-(2-{[(5-cyano-1-methyl-1$H$-pyrazol-3-yl)methyl](methyl)carbamoyl}-5-nitrophenyl)ethoxy]pyridin-2-yl}imidodicarbonate 11 (7.0 g, 9.76 mmol), KOAc (3.17 g, 32.3 mmol), Pd(OAc)$_2$ (167 mg, 0.745 mmol) and H$_2$O (176 mg, 9.80 mmol) in 2-methyl-2-butanol (210 ml) was flushed with N$_2$ for 3 min. CataCXiumA (267 mg, 0.75 mmol) was added, and the mixture was flushed with N$_2$ for a further 3 min and the yellow reaction mixture was then stirred at a reflux for 16 h. LCMS indicated that 11 had been consumed, and the desired product was formed. The yellow mixture was poured into ice (20 g) and extracted with EtOAc (3 × 250 ml). The combined organic layers were washed with brine (3 × 50 ml), dried over Na$_2$SO$_4$, filtered and concentrated to give a residue, which was diluted with EtOAc (15 ml), and subsequently filtered. The filter cake was washed with EtOAc (5 ml × 3) and dried to give di-*tert*-butyl [3-cyano-2,10,16-trimethyl-12-nitro-15-oxo-10,15,16,17-tetrahydro-2$H$-8,4-(metheno) pyrazolo [4,3-$h$][2,5,11] benzoxadiazacyclotetradecin-7-yl]imidodicarbonate 4 (4.0 g, 79%) as a grey solid.

**7-Amino-2,10,16-trimethyl-12-nitro-15-oxo-10,15,16,17-tetrahydro-2H-8, 4-(metheno)pyrazolo [4,3-h][2,5,11]benzoxadiazacyclotetradecine-3-carbonitrile 5.** To a stirred solution of di-*tert*-butyl [3-cyano-2,10,16-trimethyl-12-nitro-15-oxo-10,15,16,17-tetrahydro-2$H$-8,4-(metheno)pyrazolo [4,3-$h$][2,5,11] benzoxadiazacyclotetradecin-7-yl]imidodicarbonate 4 (4.0 g, 6.31 mmol) in EtOAc (30 ml) was added HCl(g)/EtOAc (4 M, 30 ml) at 0 °C. After the addition was completed, the light-yellow solution was allowed to slowly warm to 20 °C and stirred for 16 h. LCMS showed that the starting material still remained, and as such a further portion of HCl(g)/EtOAc (4 M, 20 ml) was added. The suspension was stirred at 20 °C for 20 h before a further portion of HCl(g)/EtOAc (4 M, 20 ml) was added. The suspension was again stirred at 20 °C for 20 h, after which time LCMS showed that the starting material had been consumed with a new major peak being the desired product. The suspension was concentrated, and the residue diluted with H$_2$O (20 ml), and basified with saturated NaHCO$_3$ solution to pH $\sim$8. The suspension was filtered with the filtrate being discarded. The cake was eluted with EtOAc until no further product could be detected in the filtrate. The organic layer was collected, dried over Na$_2$SO$_4$ and concentrated to give 7-amino-2,10,16-trimethyl-12-nitro-15-oxo-10,15,16,17-tetrahydro-2$H$-8,4-(metheno)pyrazolo [4,3-$h$][2,5,11]benzoxadiazacyclo tetra decine-3-carbonitrile 5 (2.4 g, 88%) as a yellow solid. LCMS (APCI) $m/z$ 433.9 $[M + H]^+$; $^1$H NMR (400 MHz, CDCl$_3$) δ 8.48 (d, $J = 1.2$ Hz, 1 H), 8.18 (d, $J = 2.4$ Hz, 1 H), 7.84 (s, 1 H), 7.42 (d, $J = 8.4$ Hz, 1 H), 6.86 (s, 1 H), 5.77 (q, $J = 6.4$ Hz, 1 H), 4.94 (bs s, 2 H), 4.39 (q, $J = 14.0$ Hz, 2 H), 4.08 (s, 3 H), 3.17 (s, 3 H), 1.85 (d, $J = 6.4$ Hz, 3 H).

**Separation of (10R)-7-amino-2,10,16-trimethyl-12-nitro-15-oxo-10,15,16,17-tetrahydro-2H-8,4-(metheno)pyrazolo[4,3-h][2,5,11]benzoxadiazacyclote-tradecine-3-carbonitrile (R)-5 and (10S)-7-amino-2,10,16-trimethyl-12-nitro-15-oxo-10,15,16,17-tetrahydro-2H-8,4-(metheno)pyrazolo[4,3-h][2,5,11] benzoxadiazacyclotetradecine-3-carbonitrile (S)-5.** With 31 synthesized, Boc-protection and standard ester hydrolysis provides the acid 9, which is condensed with the previously reported pyrazole 10, to give the direct arylation precursor 11 (Supplementary Fig. 8). Again, direct arylation under standard conditions leads to 4, which is accompanied with minor amounts of mono-Boc-protected material as well as 5. Exhaustive acidic deprotection leads to 5 in 71% yield over two steps.

The analytical chiral separation was performed by SFC on a Chiralpak AD (100 × 4.6 mm I.D., 3 μm particle size), which was eluted with 40% ethanol (0.05% DEA) in CO$_2$. The flow rate of 2.8 ml min$^{-1}$ gave Rt$_{(Peak\ 1)}$ = 1.193 min and Rt$_{(Peak\ 2)}$ = 1.799 min. The racemic mixture of 5 (13 g) was purified by preparative SFC (Chiralpak AD 250 × 50 mm I.D., 10 μm particle size, which was eluted with 45% ethanol with 0.05% NH$_4$OH at a flow rate of 200 ml min$^{-1}$) and gave peak 1 ((R)-5) as a yellow solid (4.94 g, 38%, $ee$ = 100%) and peak 2 ((S)-5) as a yellow solid (6.01 g, 46%, $ee$ = 98%).

(10R)-7-amino-2,10,16-trimethyl-12-nitro-15-oxo-10,15,16,17-tetrahydro-2H-8,4-(metheno)pyrazolo[4,3-h][2,5,11]benzoxadiazacyclotetradecine-3-carbonitrile (R)-5 (Peak 1): 100% ee. LCMS (APCI) m/z 434.0 [M+H]$^+$; $^1$H NMR (600 MHz, CDCl$_3$) δ 8.47 (d, J = 2.3 Hz, 1 H), 8.15 (dd, J = 8.4, 2.3 Hz, 1 H), 7.81 (d, J = 1.8 Hz, 1 H), 7.41 (d, J = 8.4 Hz, 1 H), 6.85 (d, J = 1.8 Hz, 1 H), 5.76 (q, J = 6.3 Hz, 1 H), 4.94 (bs s, 2 H), 4.39 (q, J = 14.0 Hz, 2 H), 4.07 (s, 3 H), 3.16 (s, 3 H), 1.84 (d, J = 6.3 Hz, 3 H); $^{13}$C NMR (151 MHz, CDCl$_3$) δ 167.82, 149.77 (d, J = 4.9 Hz), 149.39, 143.66, 142.93, 141.53, 138.88 (d, J = 2.6 Hz), 137.61, 127.51, 127.40, 123.09, 123.05, 118.39, 114.59, 112.61, 110.60, 71.40, 47.52, 38.90, 31.64, 22.60. Correlates to material synthesized from (S)-27.

(10S)-7-amino-2,10,16-trimethyl-12-nitro-15-oxo-10,15,16,17-tetrahydro-2H-8,4-(metheno)pyrazolo[4,3-h][2,5,11]benzoxadiazacyclotetradecine-3-carbonitrile (S)-5 (Peak 2): 98% ee. LCMS (APCI) m/z 433.9 [M+H]$^+$.

The synthesis of di-tert-butyl [(10R)-3-cyano-12-iodo-2,10,16-trimethyl-15-oxo-10,15,16,17-tetrahydro-2H-8,4-(metheno)pyrazolo[4,3-h][2,5,11]benzoxadiazacyclotetradecin-7-yl]imidodicarbonate (R)-6 (Supplementary Fig. 9) is described below.

## Di-tert-butyl [(10R)-3-cyano-2,10,16-trimethyl-12-nitro-15-oxo-10,15,16,17-tetrahydro-2H-8,4-(metheno)pyrazolo[4,3-h][2,5,11]benzoxadiazacyclotetradecin-7-yl]imidodicarbonate (R)-4.

A mixture of (10R)-7-amino-2,10,16-trimethyl-12-nitro-15-oxo-10,15,16,17-tetrahydro-2H-8,4-(metheno)pyrazolo[4,3-h][2,5,11]benzoxadiazacyclotetradecine-3-carbonitrile (R)-5 (600 mg, 1.38 mmol), (Boc)$_2$O (3.02 g, 13.8 mmol) and DMAP (84.6 mg, 0.692 mmol) in CH$_2$Cl$_2$ (40 ml) was stirred at 25 °C for 16 h. TLC (CH$_2$Cl$_2$/MeOH = 20:1) indicated that the starting material had been consumed and a major new spot detected. The reaction was concentrated in vacuo to give the crude product, which was purified by silica gel chromatography (CH$_2$Cl$_2$/MeOH = 20:1), which was further purified by preparative TLC (CH$_2$Cl$_2$/MeOH = 20:1, R$_f$~0.6) to give di-tert-butyl [(10R)-3-cyano-2,10,16-trimethyl-12-nitro-15-oxo-10,15,16,17-tetrahydro-2H-8,4-(metheno)pyrazolo[4,3-h][2,5,11]benzoxadiazacyclotetradecin-7-yl]imidodicarbonate (R)-4 (670 mg, 76%) as a white solid. LCMS (APCI) m/z 656.0 [M+Na]$^+$. $^1$H NMR (600 MHz, CDCl$_3$) δ 8.50 (d, J = 2.2 Hz, 1 H), 8.25 (d, J = 1.9 Hz, 1 H), 8.16 (dd, J = 8.4, 2.3 Hz, 1 H), 7.39 (d, J = 8.4 Hz, 1 H), 7.19 (d, J = 1.9 Hz, 1 H), 5.78 (q, J = 6.3 Hz, 1 H), 4.51–4.34 (q, J = 14.03, 20.85 Hz, 2 H), 4.12 (s, 3 H), 3.18 (s, 3 H), 1.82 (d, J = 6.3 Hz, 3 H), 1.51 (s, 19 H); $^{13}$C NMR (151 MHz, CDCl$_3$) δ 167.88, 151.18, 149.39, 148.29, 144.54, 142.04, 141.92, 141.17, 137.90, 127.12, 125.73, 125.46, 124.20, 123.31, 121.37, 113.81, 110.06, 83.58, 71.94, 47.52, 39.14, 31.66, 28.13, 22.85.

## Di-tert-butyl [(10R)-12-amino-3-cyano-2,10,16-trimethyl-15-oxo-10,15,16,17-tetrahydro-2H-8,4-(metheno)pyrazolo[4,3-h][2,5,11]benzoxadiazacyclotetradecin-7-yl]imidodicarbonate (R)-12.

A solution of Na$_2$S$_2$O$_4$ (165 mg, 0.947 mmol) and NaHCO$_3$ (79.5 mg, 0.947 mmol) in H$_2$O (10 ml) was added in a dropwise manner to a mixture of di-tert-butyl [(10R)-3-cyano-2,10,16-trimethyl-12-nitro-15-oxo-10,15,16,17-tetrahydro-2H-8,4-(metheno)pyrazolo[4,3-h][2,5,11]benzoxadiaza cyclotetradecin-7-yl]imidodicarbonate (R)-4 (150 mg, 0.237 mmol) in THF (10 ml) at 0 °C. The mixture was stirred at 20 °C for 16 h, at which point LCMS showed that ~ 60% of the starting material remained. A solution of Na$_2$S$_2$O$_4$ (165 mg, 0.947 mmol) and NaHCO$_3$ (79.5 mg, 0.947 mmol) in H$_2$O (5 ml) was again added in a dropwise manner to the mixture at 0 °C. The mixture was stirred at 20 °C for 16 h. The mixture was extracted with EtOAc (30 ml), and the organic layer washed with brine (10 ml), dried over Na$_2$SO$_4$ and concentrated under vacuum to give the crude product as a yellow solid. The crude product was further purified by preparative TLC (CH$_2$Cl$_2$/MeOH = 15:1, R$_f$~0.5) to give di-tert-butyl [(10R)-12-amino-3-cyano-2,10,16-trimethyl-15-oxo-10,15,16,17-tetrahydro-2H-8,4-(metheno)pyrazolo[4,3-h][2,5,11]benzoxadiazacyclotetradecin-7-yl]imidodicarbonate (R)-12 (83 mg, 58%) as a white solid. LCMS (APCI) m/z 588.9 [M+H]$^+$; $^1$H NMR (400 MHz, CDCl$_3$) δ 8.25–8.23 (m, 1 H), 7.21 (s, 1 H), 6.96 (d, J = 5.2 Hz, 1 H), 6.71 (s, 1 H), 6.53–6.51 (m, 1 H), 5.74–5.72 (m, 1 H), 4.61–4.59 (m, 1 H), 4.39–4.37 (m, 1 H), 4.11 (s, 3 H), 3.12 (s, 3 H), 1.75 (d, J = 6.4 Hz, 3 H), 1.53 (br s, 2 H), 1.47 (s, 18 H).

## Di-tert-butyl [(10R)-3-cyano-12-iodo-2,10,16-trimethyl-15-oxo-10,15,16,17-tetrahydro-2H-8,4-(metheno)pyrazolo[4,3-h][2,5,11]benzoxadiazacyclotetradecin-7-yl]imidodicarbonate (R)-6.

To a stirred solution of di-tert-butyl [(10R)-12-amino-3-cyano-2,10,16-trimethyl-15-oxo-10,15,16,17-tetrahydro-2H-8,4-(metheno)pyrazolo[4,3-h][2,5,11]benzoxadiazacyclotetradecin-7-yl]imidodicarbonate (R)-12 (230 mg, 0.381 mmol) and isoamyl nitrite (89.3 mg, 0.762 mmol) in CH$_3$CN (20 ml) was added CH$_2$I$_2$ (204 mg, 0.762 mmol) at 10 °C. The mixture was stirred at 80 °C for 4 h. TLC (CH$_2$Cl$_2$/MeOH = 20:1) indicated that no starting material remained and several new spots were detected. The reaction mixture was concentrated under vacuum to give the crude product as yellow oil, which was initially purified by preparative TLC (CH$_2$Cl$_2$/MeOH = 20:1) to give a yellow solid. This was further purified by preparative HPLC (to yield di-tert-butyl [(10R)-3-cyano-12-iodo-2,10,16-trimethyl-15-oxo-10,15,16,17-tetrahydro-2H-8,4-(metheno)pyrazolo[4,3-h][2,5,11]benzoxadiazacyclotetradecin-7-yl]imidodicarbonate (R)-6 (59 mg, 21%) as a white solid. LCMS (APCI) m/z 714.6 [M+H]$^+$; $^1$H NMR (400 MHz, CDCl$_3$) δ 8.27 (d, J = 2 Hz, 1 H), 7.93 (s, 1 H), 7.65 (d, J = 8.4 Hz, 1 H), 7.17 (s, 1 H), 6.94 (d, J = 8.0 Hz, 1 H), 5.66–5.61

(m, 1 H), 4.52–4.38 (m, 2 H), 4.12 (s, 3 H), 3.13 (s, 3 H), 1.76 (d, J = 6.4 Hz, 3 H), 1.50 (s, 18 H); $^{13}$C NMR (101 MHz, CDCl$_3$) δ 168.99, 150.84, 148.29, 144.86, 141.71, 141.17, 137.42, 137.23, 135.00, 127.22, 125.58, 125.33, 121.27, 113.49, 110.07, 96.80, 83.11, 71.87, 47.46, 38.90, 31.45, 28.06, 22.72.

## (R)-19-Amino-13-(7,9-dioxo-6,10-dioxa-8-spiro[4.5]decylideneiodo)-4,8-dimethyl-16-methyl-9-oxo-17-oxa-4.5.8.20-tetraazatetracyclo[16.3.1.0$^2$,$^6$.0$^{10,15}$]docosa-1(22),2,5,10(15),11,13,18,20-octaene-3-carbonitrile (R)-13.

To a solution of aryl iodide (R)-6 (0.1 g, 0.14 mmol) in anhydrous acetonitrile (8 ml), under an atmosphere of Ar(g) was added trimethylsilyl acetate (63 μl, 0.42 mmol), followed by Solid Selectfluor (0.1 g, 0.28 mmol). The reaction mixture was stirred at room temperature for 20 h and volatile contents were then removed by rotary evaporation. Ethanol (2 ml) was added to the residue followed by a solution of 6,10-dioxaspiro[4.5]decane-7,9-dione (0.024 g, 0.14 mmol) dissolved in aqueous Na$_2$CO$_3$ (1.5 ml, 0.4 M). The reaction mixture was vigorously stirred at room temperature for 0.5 h, until full conversion of iododiacetoxy starting materials was determined by TLC. The reaction mixture was then diluted with water (~8 ml) and extracted with CHCl$_3$ (3 × 10 ml). The combined organic extracts were dried with anhydrous Na2SO4, filtered and concentrated. The reaction mixture was purified by flash chromatography (10% MeOH in EtOAc) to yield the desired product in 36% yield (0.044 g, 0.05 mmol).

$^1$H NMR: (300.1 MHz, CDCl$_3$) δ 8.18 (d, J = 1.9 Hz, 1 H), 8.04 (d, J = 1.8 Hz, 1 H), 7.71 (dd, J = 8.4, 1.9 Hz, 1 H), 7.26 (d, J = 8.3 Hz, 1 H), 7.11 (d, J = 1.9 Hz, 1 H), 5.73 (q, J = 6.4 Hz, 1 H), 4.31 (m, J = Hz, 2 H), 4.12 (s, J = Hz, 3 H), 3.17 (s, J = Hz, 3 H), 2.19–2.14 (m, J = Hz, 4 H), 1.82–1.78 (m, J = Hz, 7 H), 1.58 (s, J = Hz, 18 H) p.p.m.; $^{13}$C NMR: (75.5 MHz, CDCl3) δ 167.30, 164.24, 151.75, 148.27, 144.61, 143.21, 141.86, 138.74, 138.46, 133.29, 131.44, 129.15, 125.75, 125.62, 123.13, 115.27, 114.30, 113.72, 109.75, 83.67, 72.74, 55.56, 47.20, 39.00, 37.39, 31.49, 29.69, 29.26, 28.18, 23.35, 22.52 p.p.m.; HRMS (m/z) [M+H]$^+$ calcd. for C$_{39}$H$_{44}$IN$_6$O$_{10}$, 883.2158 found 883.2169 and [M+Na]$^+$ calcd. for C$_{39}$H$_{43}$IN$_6$O$_{10}$Na 905.1978 found 905.1989.

**General method for radioisotope production and preparation.** One-step radiosynthesis of [$^{11}$C]lorlatinib. Following the 'loop method' on a GE Tracerlab FX$_{FN}$ radiofluorination module was cleaned and the HPLC loop cleaned and dried according to standard methods. Briefly, the methyl iodide delivery line and HPLC loop were washed with 10 ml of water, 10 ml of ethanol and 10 ml of acetone. The methyl iodide delivery line was reattached to the GE FX$_M$ module and helium purged through the system for 20 min to ensure that the HPLC loop was dried.

One milligram of the unprotected precursor (R)-2 was dissolved in 100 μl of a solution prepared from 12 μl of 1 M TBAOH in methanol dissolved in 1 ml of anhydrous DMF. Eighty millilitre of this solution was transferred to the HPLC loop just prior to the release of the activity from the cyclotron. A concentration of 0.7 equivalents of the TBAOH base was used to allow for only the partial removal of the amide proton and to insure that no other acidic protons should be removed. $^{11}$CO$_2$ (800 mCi) was converted to $^{11}$CH$_3$I using the GE FX$_M$ (gas-phase methyl iodide production module) and the carbon-11-labelled methyl iodide formed (~365 mCi) was transferred to the HPLC loop in 4 min with a slow stream of helium. Then, the $^{11}$CH$_3$I was allowed to react on the loop for a further 5 min. Semi-prep purification of the crude material is shown in Supplementary Fig. 12. This trace shows that it is not possible to separate the product [$^{11}$C](R)-1 from the starting precursor (R)-2.

Two-step radiosynthesis of [$^{11}$C]1. 1 mg of the diBOC-protected precursor (R)-3 was dissolved in 100 μl of a solution prepared from 12 μl of 1 M TBAOH dissolved in 1 ml of anhydrous DMF. A volume of 80 μl of this solution was transferred to the HPLC loop just prior to the release of the activity from the cyclotron. $^{11}$CO$_2$ (1,270 mCi) was converted to $^{11}$CH$_3$I and the methyl iodide formed (~620 mCi) was transferred to the HPLC loop in 4 min with a slow stream of helium. Then, the methyl iodide was allowed to react on the loop for a further 5 min. The crude reaction mixture was purified by semi-prep HPLC 60:40 Acetonitrile: 0.1 M ammonium formate, 5 ml min$^{-1}$, Phenomenex Luna C18, 10 μm, 10 × 250 mm. The labelled protected intermediate was collected at ~16 min and the collection was restricted to only 1 min to allow the pH in the subsequent hydrolysis step to be reproducible. The collected peak was transferred to the reaction vessel through the Red path (see Supplementary Fig. 13). The line between Valve 18 and the collection bulb was cut, and a male and female leur connection was placed there, allowing easy conversion between the standard set-up and the set-up for the two-step carbon-11-labelling reaction.

Prior to collecting the methylated intermediate, valves 13, 24 and 25 were energized and valve 11 was de-energized, once V18 was energized, 1 min of the peak was collected. Since the concentration and volume of the acid to be used in the hydrolysis are fixed (0.7 ml of 6 M HCl), only 5 ml of the HPLC solvent containing 0.1 M ammonium formate may be collected. If a larger amount is collected (1.5 min), then there was a significant amount of the protected material observed in the final product. A typical semi-prep HPLC trace is shown in Supplementary Fig. 14.

The hydrolysis was performed at 80 °C for 5 min, and then the reaction mixture was transferred back to the collection bulb. The bulb was preloaded with sodium hydroxide (4.5 ml, 1 M), sodium acetate (6 ml, 3 M) and water (6 ml). The sodium acetate was added to act as a buffer, which allowed for the reproducible

neutralization of the reaction mixture and thus the trapping of the desired product on the HLB Light Sep-Pak cartridge. The cartridge was then washed with sterile water for irrigation (10 ml) to remove any remaining salts and acetonitrile. The product was eluted from the SPE cartridge with ethanol (1 ml, USP, 200 proof) and diluted with sterile saline (9 ml, 0.9% solution).

Analysis of the HLB Sep-Pak indicated that >99% of the radioactivity was trapped on the cartridge. The trapped radioactive material was eluted with ethanol (1 ml) followed by elution with saline (9 ml, 0.9%). Final analysis of the HLB Sep-Pak indicated that <0.01% of the radioactivity remained on the cartridge. Quality control of the final product is shown in Supplementary Fig. 15.

[$^{18}$F]Fluoride production. A GE PETtrace 16.5 MeV cyclotron was used for [$^{18}$F]fluoride production by the $^{18}$O(p,n)$^{18}$F nuclear reaction to irradiate $^{18}$O-enriched water. A GE high-yield niobium target containing >97% enriched O-18 water (Isotec, Taiyo Nippon Sanso or Rotem) was bombarded with protons at integrated currents up to 65 μA to produce [$^{18}$F]fluoride. [$^{18}$F]Fluoride was delivered to a lead-shielded hot cell in $^{18}$O-enriched water by nitrogen gas pressure. [$^{18}$F]Fluoride was prepared for radiofluorination as follows: a solution of base (for example, tetraethylammonium bicarbonate, 3 mg) in acetonitrile and water (1 ml, v/v 7:3) was added to an aliquot of target water (≤1 ml) containing the appropriate amount of [$^{18}$F]fluoride in a V-shaped glass vial sealed with a teflon-lined septum. The vial was heated to 110 °C while nitrogen gas was passed through a P$_2$O$_5$-Drierite column followed by the vented vial. When no liquid was visible in the vial, it was removed from heat, anhydrous acetonitrile (1 ml) was added and the heating was resumed until dryness. This azeotropic drying step was repeated an additional three times. The vial was then cooled at room temperature under nitrogen pressure.

Manual synthesis of [$^{18}$F](R)-1 via (R)-5. Manual radiolabelling with tetraethyl ammonium [$^{18}$F]fluoride ([$^{18}$F] TEAF) was carried out following standard labelling procedures using precursor (R)-5 (3–4 mg) in DMSO and heated at 215 °C for 15 min. Aliquots of the reaction mixture were measured by radioTLC and showed no significant yield of the desired product. Subsequent analysis of the reaction mixture by radioHPLC indicated that <0.1% of the [$^{18}$F](R)-1 was being produced.

Microfluidic synthesis of [$^{18}$F]1 via (R)-4. It has been well documented that the use of microfluidic technology may lead to increased radiochemical conversion for reactions with challenging substrates, where there is loss of material due to decreased incorporation, degradation or inability to use high temperatures on a conventional system. To this end, radiosynthesis of (R)-1 was repeated and the reaction parameters optimized using a commercial microfluidic system and following the optimization procedure. The radiosynthesis was then attempted using the Discovery mode of the Advion NanoTek microfluidic system[38], and the optimal conditions were determined to be ~14 mg ml$^{-1}$ (3 mg) of the nitro precursor (R)-4 in DMSO (200 μl), a 100 μm × 2 m reactor at a temperature of 220 °C, with a flow rate of 40 μl min$^{-1}$. To ensure complete deprotection of all labelled materials, the product was hydrolysed using 1 ml of ethanolic HCl, at 100 °C for 3 min; however, complete HPLC separation of the nitro precursor from the final product proved to be problematic.

However, despite these efforts, the isolated yield of this reaction was only just over 1%. Efforts to improve the RCY by carrying out the reaction under less basic conditions by using tetrabutylammonium sulfate in lieu of tetraethylammonium carbonate yielded similar results and we focused our attention on preparing (R)-13 (Supplementary Fig. 10).

Manual labelling [$^{18}$F]1 via iodonium ylide (R)-13. Precursor (R)-13 (2 mg) was dissolved in DMF (400 μl) and added to a V-vial containing azeotropically dried [$^{18}$F]Et$_4$NF (typically 2–4 mCi). The reaction was heated at 80 °C for 10 min. The reaction mixture was cooled for 3 min and then HCl was added (4 M, 1 ml) and the reaction heated at 80 °C for 10 min. The reaction mixture was cooled to room temperature and neutralized to pH 5 with the addition of a mixture of NaOH (5 M, 0.8 ml) and NaOAc (3 M, 0.2 ml). The resulting mixture was further diluted with water (16 ml) and passed through a Waters C18 Sep-Pak, which had been activated by flushing sequentially with ethanol (1 ml) and water (5 ml). The Sep-Pak was flushed with water (2 ml) and the desired product was eluted with ethanol (1 ml). Product identity and purity were determined by radioHPLC (30:70 CH$_3$CN:H$_2$O + 0.1 N ammonium formate, Waters X-Bridge phenyl column) and radioTLC (EtOAc + 5% EtOH; Supplementary Fig. 19). The product was >85% radiochemically pure. RCY was determined as the percentage of radioactivity that was isolated as the final product from the amount of activity present in the V-vial before the addition of (R)-13 to dried [$^{18}$F]Et$_4$NF, and is not decay-corrected. Using SPE only for the purification of the final product, [$^{18}$F]lorlatinib (R)-1 is not sufficient to purify the radiotracer as there is a radiochemical by-product that is not removed during the C18-SPE purification, as well as a number of cold impurities as evidenced by the ultraviolet HPLC trace.

Purification of (R)-8 on the GE TracerLab FX$_{FN}$. Precursor (R)-13 (2 mg) was dissolved in DMF (400 μl) and added to a V-vial containing azeotropically dried [$^{18}$F]Et$_4$NF (24.7 mCi). The reaction was heated at 80 °C for 10 min. The reaction mixture was cooled at 0 °C for 1 min and then diluted with water (2 ml). The crude reaction mixture (19 mCi) was placed into RV2 of FX$_{FN}$ synthesis module and manually loaded onto HPLC equipped with a Phenomenex Luna C18 5 u Semi-Prep column that was previously equilibrated with 60:40 MeCN/0.1 M ammonium formate. The desired compound eluted ~17 min at 5 ml min$^{-1}$ HPLC flow and was collected in a bulk vessel with water (25 ml). The contents of the bulk vessel were passed through a C18-SPE cartridge that has been previously installed

on the FX$_{FN}$ module. The SPE cartridge was washed with water (10 ml) and taken off line. The total activity on Sep-Pak = 4.64 mCi. A volume of 1 ml of EtOH was used to elute product into a clean V-vial −4.64 mCi, RCY = 19% non-decay-corrected. To the contents of the v-vial was added HCl (4 M, 1 ml) and the reaction heated at 90 °C for 10 min. The reaction was cooled to room temperature and neutralized to pH 5 with the addition of a mixture of NaOH (5 M, 0.8 ml) and NaOAc (3 M, 0.5 ml). The contents of the vial was loaded onto a prepared HLB cartridge (3.55 mCi) and [$^{18}$F]Lorlatinib (R)-1 was eluted with EtOH (1 ml) −3.36 mCi in a RCY of 14% (non-decay corrected). Radiochemical purity = >97%; specific activity = 101 mCi μmol$^{-1}$.

Automation of [$^{18}$F]lorlatinib. After all reagents and materials were installed on the GE TracerLab FX$_{FN}$, the [$^{18}$F]fluoride, 7 mCi, was transferred to the GE TracerLab FX$_{FN}$ and the automated sequence was initiated. After completing the HPLC purification of intermediate (R)-8, the compound was eluted from the Sep-Pak using 1 ml of ethanol into a clean V-vial −3.5 mCi. To the contents of the V-vial was added HCl (4 M, 1 ml) and the reaction heated at 90 °C for 10 min. The reaction was cooled to room temperature and neutralized to pH 5 with the addition of a mixture of NaOH (5 M, 0.8 ml) and NaOAc (3 M, 0.5 ml). The contents of the vial were loaded onto a prepared HLB cartridge (3.55 mCi) and [$^{18}$F]lorlatinib (R)-1 was eluted with EtOH (1 ml) to give 1.22 mCi at 84 min of [$^{18}$F]lorlatinib (R)-1, which was >97% radiochemically pure.

Animal husbandry. All animal experiments for both rodents and NHPs were conducted in compliance with the Institutional Animal Care and Use Committee (IACUC) guidelines and the Guide for the Care and Use of Laboratory Animals.

PET image acquisition. All experiments were performed in compliance with the procedures authorized by the Institutional Animal Care and Use Committee at Massachusetts. Studies were performed on two adult male rhesus macaques (~10 kg body weight). Animals were sedated with ketamine (10 mg kg$^{-1}$, intramuscularly) and transported from housing to the imaging suite where anaesthesia was maintained by isoflurane (1–2% in 100% O$_2$ at 2 l min$^{-1}$). A flexible catheter was placed in the saphenous vein for administration of radiotracer and the animal was positioned in the bore of an ECAT EXACT HR+ PET scanner. Transmission data were acquired for 210 s at each of six consecutive bed positions, with location and overlap corresponding to those used for subsequent emission data collection and with field of view covering the entire head and spanning down approximately to the knees. Approximately 4.5 mCi of [$^{11}$C]lorlatinib prepared at high specific activity was infused intravenously over 60 s followed by flush with sterile saline. Collection of emission data was initiated concurrent with the start of administration of radiotracer. The dynamic multibed acquisition consisted of six bed positions (matching the transmission scan described above) and 11 passes. Emission data were acquired in a three-dimensional mode for 60 s at each bed position during the first eight passes and 180 s per bed position for the final three passes. Accounting for latencies during transition between bed positions, the emission data were collected out to 128 min with data acquisition starting at the follow times for the successive passes: 0, 8.5, 17, 25.5, 34, 42.5, 51, 59.5, 68, 88.5 and 109 min after start of scan. PET images were reconstructed on a grid with 2.57 mm voxels in plane and 2.43 mm slice thickness using the manufacturer's implementation of the filtered backprojection algorithm with corrections for photon scatter and attenuation, random coincidences, system deadtime and detector inhomogeneity. PET radioactivity concentrations were decay-corrected to the start of the scan and normalized by injected dose and subject body weight to produce images in units of SUV. Images are shown below for each animal, with data summed either prior to decay correction (displayed in units of Bq/cc) or after decay correction (in SUV units).

Data availability. The data that support the findings of this study are available from the corresponding author upon reasonable request.

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

## Acknowledgements

We would like to thank the staff at the radiochemistry programme, Division of Nuclear Medicine and Molecular Imaging, at Massachusetts General Hospital, for technical support. We thank WuXi AppTec for partial chemistry support. S.H.L. is a recipient of NIH career development award from the National Institute on Drug Abuse (DA038000) and N.V. thanks National Institute on Ageing of the NIH R01AG054473) and Pfizer Inc. for support.

## Author contributions

T.L.C., N.S., E.L., S.H.L., W.W. and P.R. contributed to the chemistry and radiochemistry. M.D.N., D.W.W., S.A.E, M.G.S. and G.E.F. contributed to the PET imaging and dosimetry study. J.C., K.M., R.N.W. and U.M. contributed to the study design. T.L.C., P.R. and N.V. prepared the manuscript and supervised the project. All the authors discussed the experimental results and commented on the manuscript.

## Additional information

**Competing interests:** The authors declare no competing financial interests.

