## [Peer review file · Nature Communications]

Reviewers' comments:

Reviewer #1 (Remarks to the Author):

The manuscript by Collier et al. constitutes an important step forward towards a routinely available ROS1/ALK radioligand for PET imaging. ROS1/ALK are crucial target kinases in the context of cancer. Especially fusion proteins have recently gained importance as a line of treatment for ALK-positive cancer patients. Imaging kinases in the brain is extremely difficult and can be considered as a hurdle that is very hard to overcome. Any effort to provide an in vivo tool to quantitatively assess the concentration of crucial kinases in brain tumor metastasis is desperately needed. The chemistry/radiochemistry described in this submission is of high quality and takes use of novel, non-traditional labeling procedures. Just recently introduced. The validation of these labeling techniques for the synthesis of complex imaging agents is very well done and supports a paradigm shift in radiolabeling. This submission is of high quality. However, some points and questions have to be clarified before publication.

1. The manuscript would improve from having less details on chemistry. A significant portion of the chemistry discussed may be placed in SI. For example Scheme 1, 2 and 4 which are highly analogous to the data previously published few years ago during the development of lorlatinib and do not add any novelty from a chemistry standpoint.

2. Was cold (R)-33 synthesized to assess the identity of the radiolabeled product (line 151)? This is important.

3. The advantages/challenges associated with the loop method versus conventional ¹¹C-methylation should be discussed. How did the conventional solution method perform in comparison to the 2-steps synthesis of ¹¹C-1? It would be important to provide this comparison.

4. A lot of details regarding radiochemistry and PET should be placed in SI or experimental section (for example details regarding products, cartridges, ethics committee, etc.).

5. Regional brain uptake of ¹¹C-lorlatinib should be provided. Is the uptake uniform. Does it match known ALK distribution? Is it expected to be non-specific?

6. At line 197 there is a claim that "The peak measured brain concentrations regionally exceeded 2 SUV at approximately 10 min post-injection (Figure 2B)." Where are the regional data? The whole brain SUV_{max} from the provided TAC seems to be about 1.6 (Fig. 2C). The reviewer could not find the regional brain data.

7. With ALK showing relatively high protein expression in the CNS, do the authors expect binding of the tracer to ALK in the normal brain? Shouldn't this be expected based on the likely suitable non-mutated ALK B_{max} in neurons, the 70 pM K_i of lorlatinib (for ALK) and good brain penetration of the tracer? It would be important to elaborate on those crucial points.

8. If there is no binding to normal ALK in the CNS, is the expectation that a radiolabeled lorlatinib tracer will enable CNS tumor visualization solely based on the anticipation that metastatic tumor cells will exhibit increase B_{max} compared to neurons due to translocation? The eventual possibility to use the ¹¹C/¹⁸F-lorlatinib tracers for CNS tumor diagnostic is the underlying theme of this paper. However, Johnson et al. (J Med Chem. 2014, 57, 4720) already demonstrated that lorlatinib is more potent towards non-mutated ALK versus most EML4-ALK mutants (for the example provided in the manuscript -line 65-, lorlatinib is about 60-fold more potent for ALK compared to EML4-ALK G1202R). If the tracer does not bind to endogenous ALK in the CNS (high affinity

target), how is it expected to bind to EML4-ALK G1202R tumor cells (60-lower potency)? Is it expected that the changes in Bmax between neurons and tumor cells (favoring tumor cells) will largely exceed the affinity discrepancies (which favor neurons)?

9. Many details from the section "Synthesis of labeling precursors for [18F]-lorlatinib" should be placed in SI.

10. At line 294, the claim that "Given that precursor (R)-4 or (R)-5 possess an electron rich aromatic ring and are therefore is not activated towards nucleophilic aromatic substitution reactions" is questionable. Such nitro precursors are mildly activated due to the presence of the para-amide and should undergo conventional fluorine-18 SNAr using routine conditions. There are multiple examples of significantly less activated aryl precursors (e.g. the case of flumazenil) which provide sufficient RCY for routine production. It seems more likely that the failure to afford the tracer in this case relate to the high temperature used for the reaction (220°C) which likely leads to precursor degradation. Under such conditions, 1% and less RCY are not surprising. Can the uv traces from the prep HPLC be provide? This would maybe confirm degradation of the precursor. Why is it that more conventional temperatures (125-170°C) were not tested to enable adequate comparison with the ylide-based radiochemistry?

11. It would be important and most interesting to provide the scan from 18F-1.

12. Many schemes with chemical structure have different templates (for example Scheme 1 vs 2 vs 3). Compounds numbers or letters in structures are sometimes bold, sometimes not (see Scheme 7), etc. Lines in Scheme 3 are not even straight. Missing 18F label in the compound number in Scheme 6. Sometimes "(a)" is used over the arrows, sometime it is "a". Please make uniform and review all figures/schemes.

Other:

- line 71 : punctuation "...neuroimaging 22-25,26,28 with..."

- Please appropriately cite: J. Med. Chem. 2015, 58, 8200–8215 as it is the only similar study available.

Reviewer #2 (Remarks to the Author):

General comments

Vasdev and co-workers present an interesting study which pushes the boundaries of conventional radiochemistry methods in the pursuit of a tracer for ALK and ROS-1. Access to such a tracer, would be of significant interest for measuring target engagement, mapping receptor expression and estimating efficacy of drugs targeting these enzymes. The authors present a number of challenging synthetic strategies to enable labelling of lorlatinib with both 11C and 18F. In vivo PET imaging studies in non-human primates indicated that lorlatinib possess reasonable BBB penetration and as such may be useful for the treatment and diagnosis of brain metastases. Overall the manuscript is of a high quality and the results are certainly novel, however the authors have not demonstrated whether or not 11C/18-F lorlatinib is a suitable tracer for these kinases. In order to be of interest for a broad audience and thus be suitable for publication in Nature Communications, the authors should present some evidence (e.g. specific binding studies, tumor uptake in ALK cancer rodent model etc) that lorlatinib may also be suitable for use as a tracer for these kinases.

Specific comments.

Page 3 – line 62: Could this also be due to a high propensity for efflux?

Page 4 – Line 74: Isn't the imaging of kinases using ATP competitive ligands even more

challenging due to do the high intracellular concentrations of ATP. Perhaps the authors can comment on this?

Page 4 – line 88: Change to to of

Change Amino- to amino-

Figure 1 – This figure could be made clearer by harmonizing the structures (ie Me or not, depiction of the macrocyclic structure). This should be done throughout the manuscript. The authors may also consider using different colours to highlight the two labelling positions.

Page 5 – lines 94-96: Is this statement necessary given that R-6 can be accessed from R-4?

Page 5 – line 97: The general reaction term nucleophilic aromatic substitution should be added here. The authors may also include a reference to 18F incorporation via this approach.

Page 5 – line 94: Please add an appropriate reference after iodonium ylide strategy

Page 6 – line 106: This synthetic strategy is very similar to that published in J. Med. Chem. 2014, 57, 4720–4744 and the authors should comment on this.

Page 6 – line 107: Isn't ring closure the only synthetic step in macrocycle formation? The author should revise this statement.

Page 6 – line 108: add reference (eg Yudin: Chem. Sci., 2015, 6, 30-49) after to address this.

Page 6 – line 113: It isn't entirely clear how the presence of a terminal methyl group would influence the precursor conformation. This difference could also be attributed to the greater nucleophilicity of secondary amines compared to their primary counterparts. Can the authors comment on this?

Page 6 – line 114: Have the authors confirmed that the cis diastereomer is the predominant form? If not, please revise this statement.

Page 6 – line 115. (R)-1 should read (R)-2

Page 6 – line 119 – dioxan should read dioxane

Page 6 – line 125 – the failure of the Boc-protection is somewhat unexpected. Can the authors comment on why this may be and shed some light on the side-products observed?

Scheme 2 and the accompanying reagents and conditions do not match (shouldn't the sequence start from (R)-7?) Please check this carefully

Page 7 – line 132: Add reference after previously reported

Page 7 – line 131: T3P is a trademark name and should read T3P®

Page 8 – line 153: The nucleophilicity of amino-pyridines is due to the resonance electron-releasing properties of the amine-substituent rather than inductive effects.

Page 8 – lines 164-173: To increase readability, the authors may consider removing the specific experimental details.

Page 9 – line 174: Please also add the decay-corrected yield

Scheme 3 – The chemical structures should be improved in this scheme. See the above remark regarding harmonization of the structures.

- As the identity of the radiolabeled product has not been determined the scheme should indicate that either (R)-1 or (R)-33 were formed

Page 12 – line 247: This statement is difficult to follow – How does the correlation of stereochemistry in an alcohol precursor lead to the postulation that 4 and 5 are good candidates for chiral separation? Please consider rewording this section.

Page 12 – line 259: The yield of the pure enantiomer should be determined based on the total amount of racemic material rather than the content of the (R)-enantiomer. The ee or er should also be given here

Page 13 – line 269: were investigated should read were also investigated

Page 14 – line 274: on the non-activated should read on non-activated

Page 14 – line 284: was the enantiopurity of the labelling precursors confirmed? Please comment on this.

Page 14 – lines 294-295: Is this really the case? The aromatic ring reacting in the S_NAr reaction is electron-deficient rather than electron-rich. It is also unclear whether or not the reaction using the Nantek system led to product formation or not – on the next page a 1% RCY is mentioned. This section should be clarified.

Given that the synthesis of 18F-(R)-1 was successful it is somewhat surprising that no pre-clinical

studies were carried out using this compound. The authors should include additional imaging experiments, at least in rodents, comparing the ¹¹C and ¹⁸F compounds. This would provide additional support that the brain uptake seen in figure 2 is due to the tracer rather than a metabolite.

Reviewer #3 (Remarks to the Author):

A well written paper. Only a few comments.

Line 71 should read This lack of effort...

Nomenclature: the only correct nomenclature for radiotracer is [¹¹C]lorlatinib, with the number superscript.. No hyphen between closing bracket and compound name.

The authors use the term SUV but do not define it. Many readers may not be familiar with it. For readers not in imaging field, giving the approx %dose that gets in the brain would be very useful.

Line 294. Not quite true that aryl ring of cmpd 5 is not activated as there is a para-positioned electron withdrawing group. Maybe is poorly activated. The method they chose is however much better.

Reviewers' comments:

Reviewer #1 (Remarks to the Author):

The manuscript by Collier et al. constitutes an important step forward towards a routinely available ROS1/ALK radioligand for PET imaging. ROS1/ALK are crucial target kinases in the context of cancer. Especially fusion proteins have recently gained importance as a line of treatment for ALK-positive cancer patients. Imaging kinases in the brain is extremely difficult and can be considered as a hurdle that is very hard to overcome. Any effort to provide an in vivo tool to quantitatively assess the concentration of crucial kinases in brain tumor metastasis is desperately needed. The chemistry/radiochemistry described in this submission is of high quality and takes use of novel, non-traditional labeling procedures. Just recently introduced. The validation of these labeling techniques for the synthesis of complex imaging agents is very well done and supports a paradigm shift in radiolabeling. This submission is of high quality. However, some points and questions have to be clarified before publication.

We greatly appreciate the Reviewer's support of our manuscript. We have addressed all of the points suggested below:

1. The manuscript would improve from having less details on chemistry. A significant portion of the chemistry discussed may be placed in SI. For example Scheme 1, 2 and 4 which are highly analogous to the data previously published few years ago during the development of lorlatinib and do not add any novelty from a chemistry standpoint.

We have now omitted Scheme 1, Scheme 2 and Scheme 4 from the manuscript. Related text has been moved to Supporting Information (SI).

2. Was cold (R)-33 synthesized to assess the identity of the radiolabeled product (line 151)? This is important.

Cold (R)-33 was not synthesized to assess the identity of the radiolabeled product because it is a salt and would have had a significantly different retention time. We employed a 2-step method which uses a diBoc-protected amine for routine production. We agree that this point is important and have clarified this with the following sentence:

“In light of the challenge to isolate [¹¹C](R)-1 from the reaction mixture with confidence and suitable radiochemical purity for clinical translation, we abandoned the 1-step strategy and focused on a more robust 2-step ¹¹C-labelling process that employs a diBoc-protected amine for routine radiotracer production.”

3. The advantages/challenges associated with the loop method versus conventional 11C-methylation should be discussed. How did the conventional solution method perform in comparison to the 2-steps synthesis of 11C-1? It would be important to provide this comparison.

Due to the known advantages of the loop method over traditional vial-based methods, and the establishment of this technique in routine ¹¹C-radiopharmaceutical production, our laboratory does not

carry out any routine ^{11}C -radiopharmaceutical production in a vial. Therefore, no attempts were made to carry out ^{11}C -methylation in a vial on this compound. We have clarified the advantages of using the “Loop Method” with the additional 2 sentences below:

“In this established method, the entire reaction is carried out on an HPLC loop prior to purification, and no losses to transfer of reagents occur. The “Loop Method” is commonly used instead of vial-based ^{11}C -methylation reactions, because of its simplicity (no heating or cooling required) as well as ease of automation, including adaptation to the commercial radiosynthesis modules.”

4. A lot of details regarding radiochemistry and PET should be placed in SI or experimental section (for example details regarding products, cartridges, ethics committee, etc.).

We omitted the text pertaining to solid phase extraction cartridges and purification details. We have also moved the description of the microfluidic radiofluorination from the main document to the SI. We have omitted the Ethics Committee Statement from the main document and it is only in the SI.

5. Regional brain uptake of ^{11}C -lorlatinib should be provided. Is the uptake uniform. Does it match known ALK distribution? Is it expected to be non-specific?

We have now included regional brain uptake for ^{11}C -lorlatinib, specifically including regions of the thalamus, frontal cortex, cerebellum, and white matter. Regions of expected ALK distribution include: olfactory bulb, cerebellum, throughout the cortical gray matter with strong expression in frontal and entorhinal cortices, and subcortical regions including the thalamus and hypothalamus¹. Many of these regions are very small and the PET scanner spatial resolution limited our ability to properly examine uptake in these smaller regions. We did find increased uptake in the cerebellum and relatively lower uptake in the white matter. Throughout the rest of the brain, uptake was generally quite uniform throughout, however, these are healthy primates and expression is expected throughout the cortex and the spatial resolution of the scanner used for these whole body imaging studies is not amenable to delineation of fine anatomical detail. Furthermore, clearance of ^{11}C -lorlatinib is generally rapid with no indication of high nonspecific uptake.

We have also added the following text to describe the regional distribution:

“The peak measured brain concentrations were locally high with the cerebellum exceeding a standardized uptake value (SUV) of 2 at approximately 10 min post-injection (Figure 2B). Additionally, the maximum measured percent injected dose in the brain was 1.4% at approximately 10 minutes post-injection. SUV time activity curves exhibit rapid uptake followed by fast washout of the radiotracer in normal non-human primates (Figure 2C). Regional uptake exhibits modest heterogeneity but is generally concordant with expected ALK distribution, with highest [^{11}C]lorlatinib concentrations in cerebellum, frontal cortex, and thalamus; intermediate levels in other cortical gray matter; and lowest values in white matter (Bilsland et al, 2008).”

Furthermore, a revised figure with regional distribution is shown below:

6. At line 197 there is a claim that “The peak measured brain concentrations regionally exceeded 2 SUV at approximately 10 min post-injection (Figure 2B).” Where are the regional data? The whole brain SUVmax from the provided TAC seems to be about 1.6 (Fig. 2C). The reviewer could not find the regional brain data.

We have now included the regional brain data (see above). Additionally, we have edited this point in the manuscript:

“The peak measured brain concentrations were locally high with the cerebellum exceeding a standardized uptake value (SUV) of 2 at approximately 10 min post-injection (Figure 2B).”

7. With ALK showing relatively high protein expression in the CNS, do the authors expect binding of the tracer to ALK in the normal brain? Shouldn't this be expected based on the likely suitable non-mutated ALK Bmax in neurons, the 70 pM Ki of lorlatinib (for ALK) and good brain penetration of the tracer? It would be important to elaborate on those crucial points.

Our study was not designed to specifically quantify uptake of this radiotracer in normal brain tissues, however, our recently added regional analysis with ¹¹C-lorlatinib reflects the known ALK distribution in normal brain tissues. Based on our further analysis of the data which is now included in this revised manuscript, we agree with the Reviewer that it is reasonable to expect binding of the tracer to ALK in the normal brain. We have now elaborated on this, as stated above, with the following sentence:

“Regional uptake exhibits modest heterogeneity but is generally concordant with expected ALK distribution, with highest [¹¹C]lorlatinib concentrations in cerebellum, frontal cortex, and thalamus; intermediate levels in other cortical gray matter; and lowest values in white matter (Bilsland et al, 2008).”

Our future work will require a full pharmacokinetic analysis with blocking/displacement studies, arterial sampling in nonhuman primates.

8. If there is no binding to normal ALK in the CNS, is the expectation that a radiolabeled lorlatinib tracer will enable CNS tumor visualization solely based on the anticipation that metastatic tumor cells will exhibit increase Bmax compared to neurons due to translocation? The eventual possibility to use the ¹¹C/¹⁸F-lorlatinib tracers for CNS tumor diagnostic is the underlying theme of this paper. However, Johnson et al. (J Med Chem. 2014, 57, 4720) already demonstrated that lorlatinib is more potent towards non-mutated ALK versus most EML4-ALK mutants (for the example provided in the manuscript -line 65-, lorlatinib is about 60-fold more potent for ALK compared to EML4-ALK G1202R). If the tracer does not bind to endogeneous ALK in the CNS (high affinity target), how is it expected to bind to EML4-ALK G1202R tumor cells (60-lower potency)? Is it expected that the changes in Bmax between neurons and tumor cells (favoring tumor cells) will largely exceed the affinity discrepancies (which favor neurons)?

The uptake of the radiotracer appears to reflect normal ALK distribution in the CNS. Our ultimate use for this PET tracer is not to detect ALK-positive tumors (of any form), but to characterize them *in vivo* to guide therapies. Being a kinase inhibitor, the compound inhibits the regular protein, the EML4-ALK fusion protein as well as the mutants as pointed out to differing degrees. Although the compound is as the Reviewer points out 60 fold less potent towards the GR1202R mutant, it is important to note that this is a relative assessment, and lorlatinib is quite potent (77nM) against the selected mutant. To put this in perspective, this is more potent than crizotinib is against wild type EML4-ALK. Given this, it should be possible to visualize both EML4-ALK as well as the various mutants with possibly the biggest challenge in tumor visualization being differentiating the translocated protein from endogenous ALK. Ultimately, we agree with the Reviewer’s assessment and expect that the changes in Bmax between neurons and tumor cells will favour tumors and will largely exceed the affinity discrepancies.

In light of the Reviewer’s comment we have now clarified our goals towards applying this radiotracer for *in vivo* quantification of ALK expression and to PET imaging by revising part of the Conclusion section to read as follows:

“Carbon-11 and fluorine-18 labelled lorlatinib were prepared in good radiochemical yields and purities, *via* a unique and fully automated 2-step ¹¹C-labeling strategy, as well as our iodonium ylide-based radiofluorination methodology. The initial PET imaging study was designed to confirm that [¹¹C]lorlatinib readily crosses the blood-brain barrier. Our future work with [¹¹C]lorlatinib includes further PET imaging in rodent tumor models, normal NHPs, with concurrent automation and preclinical translation of ¹⁸F-lorlatinib.”

We have also omitted the statement that our ¹⁸F-labeled probe is targeted for ALK quantification and the final sentence in our Discussion section now reads as follows:

“This development of [¹⁸F]lorlatinib is also worthy of further investigations *in vivo*.”

9. Many details from the section "Synthesis of labeling precursors for [¹⁸F]-lorlatinib" should be placed in SI.

Several details in this section have now been omitted from the main document and moved to the SI.

10. At line 294, the claim that "Given that precursor (R)-4 or (R)-5 possess an electron rich aromatic ring and are therefore is not activated towards nucleophilic aromatic substitution reactions" is questionable. Such nitro precursors are mildly activated due to the presence of the *para*-amide and should undergo conventional fluorine-18 S_NAr using routine conditions. There are multiple examples of significantly less activated aryl precursors (e.g. the case of flumazenil) which provide sufficient RCY for routine production. It seems more likely that the failure to afford the tracer in this case relate to the high temperature used for the reaction (220 °C) which likely leads to precursor degradation. Under such conditions, 1% and less RCY are not surprising. Can the uv traces from the prep HPLC be provide? This would maybe confirm degradation of the precursor. Why is it that more conventional temperatures (125-170 °C) were not tested to enable adequate comparison with the ylide-based radiochemistry?

We agree with the Reviewer's assessment that the precursors are mildly activated by the *N*-methyl amide and have reworded this section as follows:

"Given that precursor **(R)-4 or (R)-5** possess an electron rich aromatic ring and are only mildly activated towards nucleophilic aromatic substitution reactions due to the presence of the *para*-amide, it represents a challenging substrate for labeling with fluorine-18 using conventional labeling methods. It is therefore not surprising that fluorodenitration was not fruitful in our hands by manual or microfluidic approaches (see ESI)."

We apologize that we were not clear in our experimental details for the manual and microfluidic radiofluorination temperatures, as conventional temperatures were indeed explored. Because no products were observed on analytical HPLC, there were no attempts to carry out semi-preparative purification. We have now clarified that we ramped the temperatures from 160 °C up to 220 °C in the text as follows.

"Attempted manual fluorodenitration with [¹⁸F]Et₄NF ([¹⁸F]TEAF) was performed using the nitro precursors (3-4 mg) in DMSO with heating ranging from 160 °C to 215 °C for 15 minutes, however, [¹⁸F]lorlatinib could not be identified in the reaction mixture."

11. It would be important and most interesting to provide the scan from 18F-1.

Indeed we believe it would be interesting to provide a scan from [¹⁸F]-1, and we plan to follow up with a Full Paper in the future. However, these experiments are beyond the scope of the current Communication. Further automation and optimization of the radiochemistry would be required to minimize radiation exposure to our chemists and ethics approvals would need to be in place for the animal studies. Furthermore the rigorous evaluation of these tracers in rodent tumor models as well as non-human primates are planned. We have clarified our intentions for the present Communication and future work in the Discussion and Conclusions as stated above.

12. Many schemes with chemical structure have different templates (for example Scheme 1 vs 2 vs 3). Compounds numbers or letters in structures are sometimes bold, sometimes not (see Scheme 7), etc. Lines in Scheme 3 are not even straight. Missing 18F label in the compound number in Scheme 6. Sometimes

“(a)” is used over the arrows, sometime it is ‘a’. Please make uniform and review all figures/schemes.

All Schemes have now been reviewed and made uniform.

Other:

- line 71 : punctuation ‘...neuroimaging 22-25,26,28 with...’

The references have now been correctly reformatted as follows: ^{22-26,28}

- Please appropriately cite: *J. Med. Chem.* 2015, 58, 8200–8215 as it is the only similar study available.

We have added the following sentence and citation as a new reference as follows:

“We anticipate that this methodology will be applicable for ¹⁸F-labeling of other mildly activated aromatic clinical candidates (i.e. AZD3759,³⁹ etc.).”

Reviewer #2 (Remarks to the Author):

General comments

*Vasdev and co-workers present an interesting study which pushes the boundaries of conventional radiochemistry methods in the pursuit of a tracer for ALK and ROS-1. Access to such a tracer, would be of significant interest for measuring target engagement, mapping receptor expression and estimating efficacy of drugs targeting these enzymes. The authors present a number of challenging synthetic strategies to enable labelling of lorlatinib with both ¹¹C and ¹⁸F. In vivo PET imaging studies in non-human primates indicated that lorlatinib possess reasonable BBB penetration and as such may be useful for the treatment and diagnosis of brain metastases. Overall the manuscript is of a high quality and the results are certainly novel, however the authors have not demonstrated whether or not ¹¹C/¹⁸F lorlatinib is a suitable tracer for these kinases. In order to be of interest for a broad audience and thus be suitable for publication in *Nature Communications*, the authors should present some evidence (e.g. specific binding studies, tumor uptake in ALK cancer rodent model etc) that lorlatinib may also be suitable for use as a tracer for these kinases.*

We thank the Reviewer for appreciating the novelty, potential impact and challenges to radiolabel Lorlatinib. Indeed we believe it would be interesting to provide rodent data, and we plan to follow up with a Full Paper in the future, as we have clarified in our Conclusion.

Specific comments.

Page 3 – line 62: Could this also be due to a high propensity for efflux?

The Reviewer raises a good point; we have acknowledged this in the manuscript as follows:

“A common site of metastases in non-small cell lung cancer (NSCLC) patients is in the brain where previous generation ALK-inhibitors have limited effectiveness, and may be attributed to poor blood-brain barrier (BBB) penetration or active transport out of the brain by efflux pumps.”

Page 4 – Line 74: Isn't the imaging of kinases using ATP competitive ligands even more challenging due to do the high intracellular concentrations of ATP. Perhaps the authors can comment on this?

The Reviewer raises another good point. High intracellular levels of ATP will lead to a depression in the overall specific uptake signal and, thus, adding another challenge to imaging these difficult pathways. We have supplemented the text in the manuscript:

“This lack of effort is partially attributed to the additional challenges of imaging intracellular targets, compared to high density receptor and enzyme targets at the extracellular domain, and competition at binding sites with high intracellular levels of ATP. These difficulties are further exacerbated by the challenging radiochemistry required to prepare isotopologues of the structurally complex potent and selective TKIs.”

Page 4 – line 88: Change to to of

Change made.

Change Amino- to amino-

Change made.

Figure 1 – This figure could be made clearer by harmonizing the structures (ie Me or not, depiction of the macrocyclic structure). This should be done throughout the manuscript. The authors may also consider using different colours to highlight the two labelling positions.

Figure 1 has been revised and different colours have been used to highlight the 2 labeling positions. See below:

Page 5 – lines 94-96: *Is this statement necessary given that R-6 can be accessed from R-4?*

The sentence in question has now been omitted.

Page 5 – line 97: *The general reaction term nucleophilic aromatic substitution should be added here. The authors may also include a reference to ¹⁸F incorporation via this approach.*

This sentence has now been updated as suggested (see below); no further references were added as the radiofluorination references are provided in the respective sections:

“Compounds **(R)-4** and **(R)-5** were attractive from the perspective of utilizing nucleophilic aromatic substitution approach to access the ¹⁸F-labelled material.”

Page 5 – line 94: *Please add an appropriate reference after iodonium ylide strategy*

3 references have been added.

Page 6 – line 106: *This synthetic strategy is very similar to that published in J. Med. Chem. 2014, 57, 4720–4744 and the authors should comment on this.*

All of the similar syntheses (Schemes 1, 2 and 4 and related text) in question have now been moved to Supporting Information.

Page 6 – line 107: Isn't ring closure the only synthetic step in macrocycle formation? The author should revise this statement.

This has section has been moved to the ESI and corrected as follows:

“Ring-closure is known to be the challenging step in the formation of macrocycles”

Page 6 – line 108: add reference (eg Yudin: Chem. Sci., 2015, 6, 30-49) after to address this.

This has section has been deleted from the main document and moved to the SI section. The reference by Yudin has been added as reference 1 in the SI.

Page 6 – line 113: It isn't entirely clear how the presence of a terminal methyl group would influence the precursor conformation. This difference could also be attributed to the greater nucleophilicity of secondary amines compared to their primary counterparts. Can the authors comment on this?

This comment is an extension of that made in the original paper, which stresses the importance of the chiral center in the molecule in negating the issue of atropisomers, as well as pushing the two ends of the molecule (shown by modelling) into close proximity. You can see this in the crystal structures of the acyclic precursor. Herein, the Reviewer makes a reasonable point about nucleophilicity, and we have added an addendum to the text to clarify the enhanced nucleophilicity of the secondary amine, as follows in the SI:

“The conformational preferences of the precursor also have a strong influence on the success of the ring formation,¹ and herein a modest yield was obtained after EDCI-mediated ring formation (it is interesting to contrast this with the yields obtained in the case of the clinical candidate, (**R**)-2, which differs by an additional methyl group on the amide nitrogen, which leads to a 10-20% increase in yield for the cyclization step (probably due to the greater nucleophilicity of the secondary amine). Other amide-bond forming reagents were not tested in this step, and the presence of cis/trans diastereomers (1/3) around the amide bond can clearly be observed in the NMR spectroscopic and LC-MS analysis of (**R**)-2.”

Page 6 – line 114: Have the authors confirmed that the cis diastereomer is the predominant form? If not, please revise this statement.

We appreciate that the Reviewer caught this error as the “trans” is actually the more stable diastereomer. It's correct in the experimental section but incorrect in our text. We have now corrected it. For the Reviewer's information this statement is made based on molecular modelling; we ran OPLS2001 calculations on the two conformations, and the trans is more stable by 1kcal/mol (see below). We have a fully detailed NMR report (including 2D studies) which supports this conclusion.

Page 6 – line 115. (R)-1 should read (R)-2

This has now been moved to SI and corrected.

Page 6 – line 119 – dioxan should read dioxane

This Scheme caption has been deleted.

Page 6 – line 125 – the failure of the Boc-protection is somewhat unexpected. Can the authors comment on why this may be and shed some light on the side-products observed?

We have now clarified the details regarding the Boc-protection as follows:

“Attempting to simply selectively bis-Boc-protect the aminopyridine compound (R)-2 led under standard conditions to mainly the tri-Boc protected compound, and all attempts to selectively remove the Boc-group from the amide nitrogen failed. With these results, and the limited amount of materials in hand, an alternative approach to (R)-3 was devised.”

Scheme 2 and the accompanying reagents and conditions do not match (shouldn't the sequence start from (R)-7?) Please check this carefully

Scheme 2 has been deleted.

Page 7 – line 132: Add reference after previously reported

Reference has been added.

Page 7 – line 131: T3P is a trademark name and should read T3P®

Correction is made to Scheme 5.

Page 8 – line 153: The nucleophilicity of amino-pyridines is due to the resonance electron-releasing properties of the amine-substituent rather than inductive effects.

We have rewritten the sentence in question as suggested by the Reviewer, and as follows:

“Methylation of 2-aminopyridines occurs favorably at the pyridine nitrogen due to resonance electron-releasing properties of the amino group which leads to enhanced nucleophilicity of the pyridine³⁰.”

Page 8 – lines 164-173: To increase readability, the authors may consider removing the specific experimental details.

Several sections have been removed and moved to SI or omitted, per consistent feedback from all Reviewers.

Page 9 – line 174: Please also add the decay-corrected yield

We have now added in the decay-corrected radiochemical yield and also noted the time to EOS in the text as follows:

The final product was purified by solid phase extraction (SPE) and reformulated in 10% ethanol in saline to provide [¹¹C]lorlatinib ([¹¹C](**R**)-**1**), in 3% uncorrected radiochemical yield (17% decay-corrected) at end of synthesis (EOS; 50 min) and a high specific activity of 3 Ci/μmol, with >95% radiochemical purity.”

Scheme 3 – The chemical structures should be improved in this scheme. See the above remark regarding harmonization of the structures.

All Schemes have been revised; structures are improved and harmonized.

- As the identity of the radiolabeled product has not been determined the scheme should indicate that either (R)-1 or (R)-33 were formed

The Scheme has now been revised to state that [¹¹C](R)-1 or [¹¹C](R)-7 (formerly (R)-33) is formed; see below:

Page 12 – line 247: This statement is difficult to follow – How does the correlation of stereochemistry in an alcohol precursor lead to the postulation that 4 and 5 are good candidates for chiral separation? Please consider rewording this section.

This is a good point by the Reviewer. We have corrected the text to state that the stereochemistry was correlated to the macrocycles, and the separation stage was decided based on “scale decisions” with the selection between 4 and 5 being based on chromatographic properties. This section is now rewritten as follows:

“A logistical challenge was to determine at what point to carry out the chiral separation. Having correlated the stereochemistry of the chiral alcohol (S)-19 (see ESI) to the final macrocycles, we decided that it would be more facile in terms of scale to carry out the separation on either 4 or 5 with the final decision been based on which displayed better solubility and more efficient chiral separation.”

Page 12 – line 259: The yield of the pure enantiomer should be determined based on the total amount of racemic material rather than the content of the (R)-enantiomer. The ee or er should also be given here

The ee are included in the experimental for the separated compounds. They’re right after the compound name prior to the NMR characterization. The yield of the separation is given, and is based on the amount of racemic material used.

Page 13 – line 269: were investigated should read were also investigated

Revision made.

Page 14 – line 274: on the non-activated should read on non-activated

Revision made.

Page 14 – line 284: was the enantiopurity of the labelling precursors confirmed? Please comment on this.

The enantiopurity was based on the final samples being identical to a cold sample of lorlatinib. Herein, after separation, all the chemistry is carried out remotely to the chiral center so one would not expect any racemization to occur.

Page 14 – lines 294-295: Is this really the case? The aromatic ring reacting in the S_NAr reaction is electron-deficient rather than electron-rich. It is also unclear whether or not the reaction using the Nantek system led to product formation or not – on the next page a 1% RCY is mentioned. This section should be clarified.

We agree with the Reviewer's assessment that the precursors are only mildly activated by the *N*-methyl amide and have reworded this section as follows:

“Given that precursor **(R)-4** or **(R)-5** possess an electron rich aromatic ring and are only mildly activated towards nucleophilic aromatic substitution reactions due to the presence of the *para*-amide, it represents a challenging substrate for labeling with fluorine-18 using conventional labeling methods. It is therefore not surprising that fluordenitration was not fruitful in our hands by manual or microfluidic approaches (see SI).”

Additional details on the microfluidic system have been added for clarification (per comments by Reviewer 1) and sections have been moved to SI.

Given that the synthesis of 18F-(R)-1 was successful it is somewhat surprising that no pre-clinical studies were carried out using this compound. The authors should include additional imaging experiments, at least in rodents, comparing the 11C and 18F compounds. This would provide additional support that the brain uptake seen in figure 2 is due to the tracer rather than a metabolite.

As stated, earlier, these proposed experiments are beyond the scope of the current Communication. Ethics approvals as well as rigorous evaluation of the tracers in rodent tumor models are planned. These studies are a major undertaking as species differences, metabolite/radiometabolite profiles, effects of anesthesia, study design, etc. have to be very carefully planned in order to rigorously evaluate these probes as kinase-selective and specific PET tracers. The goal of our preliminary study was to label the compounds and confirm brain penetration in primates. We have clarified our intentions for the present Communication and future work in the Discussion and Conclusions as follows:

“Carbon-11 and fluorine-18 labelled lorlatinib were prepared in good radiochemical yields and purities, *via* a unique and fully automated 2-step ^{11}C -labeling strategy, as well as our iodonium ylide-based radiofluorination methodology. The initial PET imaging study was designed to confirm that [^{11}C]lorlatinib readily crosses the blood-brain barrier. Our future work with [^{11}C]lorlatinib includes

further PET imaging in rodent tumor models, normal NHPs, with concurrent automation and preclinical translation of ¹⁸F-lorlatinib.”

Reviewer #3 (Remarks to the Author):

A well written paper. Only a few comments.

We thank the Reviewer for their support of our paper and have addressed all of their comments below:

Line 71 should read This lack of effort...

Sentence now reads: “This lack of effort is...”

Nomenclature: the only correct nomenclature for radiotracer is [¹¹C]lorlatinib, with the number superscript.. No hyphen between closing bracket and compound name.

All text and figures/schemes have now been updated to eliminate this hyphen.

The authors use the term SUV but do not define it. Many readers may not be familiar with it. For readers not in imaging field, giving the approx %dose that gets in the brain would be very useful.

We have defined this as standardized uptake values (SUV). Also, We have added %I.D./cc to the right axis of the time course plot of ¹¹C-lorlatinib uptake. We also added whole brain %ID to the manuscript:

“Additionally, percent injected dose in the brain peaked at 1.4% approximately 10 minutes post-injection.”

Line 294. Not quite true that aryl ring of compd 5 is not activated as there is a para-positioned electron withdrawing group. Maybe is poorly activated. The method they chose is however much better.

We agree with the Reviewer’s assessment that the precursors are only mildly activated by the *N*-methyl amide and have reworded this section as follows:

“Given that precursor **(R)-4** or **(R)-5** possess an electron rich aromatic ring and are only mildly activated towards nucleophilic aromatic substitution reactions due to the presence of the *para*-amide, it represents a challenging substrate for labeling with fluorine-18 using conventional labeling methods.”

Reviewers' comments:**Reviewer #2 (Remarks to the Author):**

The authors have made significant efforts to address the various reviewer comments and the manuscript has been greatly improved. The manuscript describes some very interesting chemical approaches to enable radiolabeling of lorlatinib and preliminary PET imaging which confirmed earlier studies showing that lorlatinib enters the CNS. However, the authors have not yet demonstrated whether $^{11}\text{C}/^{18}\text{F}$ -lorlatinib is suitable for imaging of ROS1/ALK. Unfortunately, in the absence of evidence supporting the use of lorlatinib as a kinase tracer, the manuscript does not meet the stringent criteria of novelty and extreme importance required for publication in Nature Communications.

Reviewer #4 (Remarks to the Author):

Dear Authors

I found this manuscript interesting to read and consider this to be an excellent report on some exciting radiochemistry with an interesting imaging application.

I was asked to review this manuscript specifically with reference to the author response to the reviewer's comments. My responses to this are as follows: -

I agree with the reviewers 1 and 2 concerning the requirement for biology data for the F18 radiolabeled agent; without this data there seems little point in including the F18 work in the manuscript. The rebuttal by the authors has some merit, this would require substantial extra work. However, the absence of this data lends the manuscript a feeling of incompleteness.

No further comments.

Reviewers' comments:

Reviewer #2 (Remarks to the Author):

The authors have made significant efforts to address the various reviewer comments and the manuscript has been greatly improved.

We appreciate that the Reviewer acknowledges the significant revisions that we made to our manuscript in light of their suggestions, as well as the other previous Reviewers, and we also believe that our manuscript is now greatly improved as a result.

The manuscript describes some very interesting chemical approaches to enable radiolabeling of lorlatinib and preliminary PET imaging which confirmed earlier studies showing that lorlatinib enters the CNS.

We thank the Reviewer for their appreciation of the challenging multistep carbon-11 chemistry and iodonium ylide based radiofluorination that were required to make isotopologues of lorlatinib, and that the overall goal of our preliminary PET study was to label lorlatinib and confirm its brain penetration in primates.

However, the authors have not yet demonstrated whether $^{11}\text{C}/^{18}\text{F}$ -lorlatinib is suitable for imaging of ROS1/ALK. Unfortunately, in the absence of evidence supporting the use of lorlatinib as a kinase tracer, the manuscript does not meet the stringent criteria of novelty and extreme importance required for publication in Nature Communications.

We acknowledge the importance of demonstrating suitability of these radiotracers for imaging of cancers in vivo. We would like to reiterate that lorlatinib is among the most well characterized, potent and selective, ALK/ROS1 inhibitors (preclinical and clinical) ever reported, and that we have incorporated both the carbon-11 and fluorine-18 isotopologues in the native site of the molecule, as to not alter in any way the compound structure and established binding of lorlatinib to ALK/ROS1. Nonetheless, the Reviewer has raised a valid concern and in light of this, we have now carried out preliminary PET-CT imaging studies showing baseline imaging of an ALK-positive tumor line (ALK-positive lung cancer xenograft (H3122) in mouse after injection of [^{11}C]lorlatinib and blocking by co-injection of the radiotracer with authentic non-labeled compound. In this work the ALK-positive tumour can be clearly visualized under baseline and blocked conditions and the time-activity curves clearly depict that blocking can be obtained with this radiotracer.

The following paragraph has now been added to the main document under the section titled: ***In vivo* PET Imaging of [^{11}C](R)-1:**

It is noteworthy that a preliminary assessment of tumor uptake of this radiotracer was also carried out in mice bearing subcutaneous human H3122 (EML4-ALK positive) NSCLC xenografts and evaluated by PET-CT imaging in conjunction with blocking studies (see ESI). Briefly, our initial results show that the tumor uptake reached its plateau in approximately 30-60 minutes after injection of [^{11}C](R)-1 (with 2.2 – 2.37 %ID/g) and co-injection with (R)-1 resulted in significant decrease in the tumor uptake (<0.4% ID/g) during the entire imaging course of 90 minutes.

And the following changes were made to the supporting information with the following figures added:

PRELIMINARY PET-CT IMAGING WITH ^{11}C —LABELLED LORLATINIB IN A TUMOUR BEARING MOUSE MODEL OF NSCLC

Human H3122 cells (EML4-ALK positive NSCLC cells) were cultured in RPMI containing 10% fetal bovine serum, and 1% Penicillin-Streptomycin at 37°C in a humid atmosphere containing 5% CO_2 and 95% air. A mouse model of human NSCLC was generated by injection of 5×10^6 cells in 0.2 ml of 1:1 (v/v) mixture of serum-free medium and Matrigel in the subcutaneous space of the athymic nude mice (n=9) using 25-gauge needle. Observation of a bulge under the skin was a sign of successful injection. Three weeks after tumor inoculation, the tumors reached 5-7 mm in diameter. Mice were divided into two groups: a total of 6 mice received 2.42 ± 0.68 mCi of [^{11}C]lorlatinib compound and 3 mice received mixture of [^{11}C]lorlatinib (2.18 ± 0.82 mCi) and 5 mg/kg of authentic non-radioactive lorlatinib (total injected volume < 300 μl). Dynamic PET/CT imaging was performed for 85 minutes after injection of the radiotracer using preclinical imaging scanner (Triumph II, Trifoil Imaging, Inc). Image analysis was performed with AMIDE (version 1.0.5) and the standard uptake value (SUV) and percentage of injected dose per gram of tumor tissue %ID/g (Mean \pm SEM) was calculated.

The results show that the tumor uptake reached its plateau in approximately 30-60 minutes after injection of the [^{11}C]lorlatinib (with 2.2 – 2.37 %ID/g). Co-injection of [^{11}C]lorlatinib with non-radioactive lorlatinib resulted in significant decrease in the tumor uptake during the entire imaging course.

Time-activity curve of uptake into tumor bearing mouse model at baseline after injection of [¹¹C]lorlatinib (Hot) showing uptake reaching plateau over 2 SUV in the tumor vs. when co-injected with authentic lorlatinib (Hot + Cold) with uptake reaching a plateau at < 0.4 SUV over the course of 90 minutes.

Representative coronal PET-CT image showing tumor bearing mouse model at (A) baseline after injection of [¹¹C]lorlatinib vs. under blocking conditions (B) where the radiotracer was co-injected with authentic lorlatinib and a reduction in tumor uptake is observed.

Furthermore, we stated in our manuscript rigorous evaluation of the tracers in rodent tumor models are planned as part of our future work.

Reviewer #4 (Remarks to the Author):

Dear Authors

I found this manuscript interesting to read and consider this to be an excellent report on some exciting radiochemistry with an interesting imaging application. I was asked to review this manuscript specifically with reference to the author response to the reviewer's comments. My responses to this are as follows:-

I agree with the reviewers 1 and 2 concerning the requirement for biology data for the F18 radiolabeled agent; without this data there seems little point in including the F18 work in the manuscript. The rebuttal by the authors has some merit, this would require substantial extra work. However, the absence of this data lends the manuscript a feeling of incompleteness.

No further comments.

We thank the Reviewer for their assessment of our exciting radiochemistry and imaging results in nonhuman primates. In light of this comment, as well as that of Reviewer 2, biological data in mouse models are now included. We have now included baseline and blocking studies with [¹¹C]lorlatinib in a mouse xenograft model with an ALK-positive cell line to demonstrate feasibility of imaging with an isotopologue of this potent and selective drug (see above changes and new figure added to supporting information). Labeling lorlatinib with fluorine-18 in the native site on the molecule will not alter in any way the compound structure or binding to the target. We don't feel in this case that this paper should not be published in the absence of in vivo imaging with this specific ¹⁸F-isotopologue as now the proof-of-concept in tumor bearing mice has been shown with ¹¹C-lorlatinib, and as the radiosynthesis of the ¹⁸F-isotopologue is a key step towards clinical translation. Such radiofluorination is not trivial given the challenge of radiolabelling poorly activated aromatic rings in molecules of such structural complexity, and herein we showed that this can be overcome using our iodonium ylide based radiofluorination strategy which could not be achieved by conventional nucleophilic displacement of a nitro-precursor. Furthermore, we plan to rigorously study the ¹⁸F-isotopologue of lorlatinib in a panel of tumor bearing mice and non-human primates, in addition to assessing the metabolic pathways and longer imaging protocols as part of our future work.

REVIEWERS' COMMENTS:

Reviewer #2 (Remarks to the Author):

The authors have now included some preliminary evidence showing that lorlatinib has the potential to be used as a ALK/ROS1 tracer. With the inclusion of this data, the manuscript is now suitable for publication in Nature Communications.

Reviewers' comments:

Reviewer #2 (Remarks to the Author):

The authors have now included some preliminary evidence showing that lorlatinib has the potential to be used as a ALK/ROS1 tracer. With the inclusion of this data, the manuscript is now suitable for publication in Nature Communications.

We thank the Reviewer for their support of our manuscript for publication in Nature Communications.